



# Spatial and temporal changes of the ozone sensitivity in China based on satellite and ground-based observations

Wannan Wang[1,2,3], Ronald van der A[3], Jieying Ding[3], Michiel van Weele[3] and Tianhai Cheng[1]

[1]Aerospace Information Research Institute, Chinese Academy of Sciences, Beijing, 100094, China
[2]University of Chinese Academy of Sciences, Beijing, 100049, China
[3]Royal Netherlands Meteorological Institute (KNMI), De Bilt, 3730 AE, the Netherlands

*Correspondence to*: Ronald van der A (ronald.van.der.a@knmi.nl)

**Abstract.** Ground-level ozone ($O_3$) pollution has been steadily getting worse in most part of eastern China during the past five years. The non-linearity of $O_3$ formation with its precursors like nitrogen oxides ($NO_x = NO + NO_2$) and volatile organic

compounds (VOCs) are complicating effective $O_3$ abatement plans. The diagnosis from space-based observations, the ratio of formaldehyde (HCHO) columns to tropospheric $NO_2$ columns ($HCHO/NO_2$), has previously been proved to be highly consistent with our current understanding of surface $O_3$ chemistry. $HCHO/NO_2$ ratio thresholds distinguishing $O_3$ formation sensitivity depend on regions and $O_3$ chemistry interactions with aerosol. To shed more light on current the $O_3$ formation sensitivity over China, we have derived $HCHO/NO_2$ ratio thresholds by directly connecting satellite-based $HCHO/NO_2$

observations and ground-based $O_3$ measurements over the major Chinese cities in this study. We find that a VOC-limited regime occurs for $HCHO/NO_2 < 2.3$ and $NO_x$-limited regime occurs for $HCHO/NO_2 > 4.2$. The $HCHO/NO_2$ between 2.3 and 4.2 reflects the transition between the two regimes. Our method shows that the $O_3$ formation sensitivity tends to be VOC-limited over urban areas and $NO_x$-limited over rural and remote areas in China. We find that there is a shift in some cities from the VOC-limited to the transitional regime that is associated with a rapid drop of anthropogenic $NO_x$ emissions owing to the

widely-applied rigorous emission control strategies between 2016 and 2019. This detected spatial expansion of the transitional regime is supported by rising surface $O_3$ concentrations. The enhanced $O_3$ concentrations in urban areas during the COVID-19 lockdown in China indicate that a protocol with simultaneous anthropogenic $NO_x$ emissions and VOC emissions controls is essential for $O_3$ abatement plans.

## 1 Introduction

Ground-level ozone ($O_3$) is one of major air pollutants that has negative impacts on human health and can result in eye and nose irritation, respiratory disease, and lung function impairment (Jerrett et al., 2009; Khaniabadi et al., 2017; Huang et al., 2018). Tian et al. (2020b) observed increased admissions for pneumonia associated with $O_3$ exposure, especially for elderly people. In addition, it also has important impacts on climate as a greenhouse gas by absorbing thermal radiation (Fishman et al., 1979; IPCC, 2014). Photochemical tropospheric $O_3$ is formed in a nonlinear manner from $O_3$ precursors such as volatile

organic compounds (VOCs) and nitrogen oxides ($NO_x = NO + NO_2$) in the presence of sunlight (Crutzen, 1974; Jacob, 2000).



In 2008, China was found to be the largest contributor to Asian emissions of carbon monoxide (CO), $NO_x$, non-methane volatile organic carbon (NMVOC), and methane ($CH_4$) (Kurokawa et al., 2013). Because of these large emissions of anthropogenic air pollutants, the Chinese State Council released the "Air Pollution Prevention and Action Plan" (APPAP) on September 2013, which has as a key task to prevent and control air pollution in China (Cai et al., 2017). Since then, critical

emission control strategies have been carried out that are designed to reduce the concentrations of six environmental pollutants: sulfur dioxide ($SO_2$), nitrogen dioxide ($NO_2$), CO, $O_3$, and particulate matter ($PM_{2.5}$ and $PM_{10}$) (Zhang et al., 2016; Feng and Liao, 2016). During the past decade, the concentrations of many pollutants including $SO_2$, $NO_2$, CO, $PM_{2.5}$ and $PM_{10}$ have declined in most cities, however, $O_3$ concentrations showed an increasing trend (Wang et al., 2017b; Wang et al., 2019b; Zeng et al., 2019). Therefore, reducing $O_3$ concentrations has become the focus of China's next air quality control strategies (Cheng

et al., 2018).

In terms of $O_3$ concentrations, the effectiveness of emissions control strategy depends on whether the photochemical regime of $O_3$ formation is VOC-limited or $NO_x$-limited (Jin et al., 2020). In the VOC-limited (or $NO_x$-saturated) regime, VOC emission reductions reduce the chemical production of organic radicals ($RO_2$), which in turn lead to decreased cycling with $NO_x$ and consequently lower concentration of $O_3$ (Milford et al., 1989). In $NO_x$-limited (or VOC-saturated) regime, $NO_x$

emission reductions reduce $NO_2$ photolysis, which is the primary source of free oxygen atoms. Therefore, in a $NO_x$-limited regime, $NO_x$ reductions reduce ambient $O_3$. In contrast, in VOC-limited regime, $NO_x$ acts to reduce $O_3$, so a $NO_x$ decrease in emissions promotes $O_3$ production (Kleinman, 1994).

The observed photochemical indicators and observation-based models (OBM) are the most commonly used tools to diagnose the $O_3$ formation sensitivity. $O_3$ production efficiency (OPE=$\Delta O_3$/$\Delta NO_z$) and the $H_2O_2$/$NO_z$ (or $H_2O_2$/$HNO_3$) ratio

are two widely used indicators to infer the $O_3$ formation regimes (Chou et al., 2011; Ding et al., 2013). Wang et al. (2017a) concluded that lower OPE values (e.g., < 4) indicate a VOC-limited regime. In contrast, higher OPE values (e.g., >7) indicate a $NO_x$-limited regime. OPE values in the medium range (e.g., 4 < OPE < 7) mark the transition between two regimes. Another indicator of the $O_3$ formation sensitivity regime is the $H_2O_2$/$NO_z$ ratio. Hammer et al. (2002) defined that in the VOC-limited regime, lower $H_2O_2$/$NO_z$ ratios would be expected and higher $H_2O_2$/$NO_z$ ratios indicate the $NO_x$-limited regime. In the past

decade, the observed photochemical indicators have been applied to identify the $O_3$ formation sensitivity in different periods and regions of China.

The OBM combines *in-situ* field observations and chemical box modeling. It is built on widely-used chemistry mechanisms (e.g., MCM, Carbon Bond, RACM or SAPRC), and applied to the observed atmospheric conditions to simulate various atmospheric chemical processes, including the *in-situ* $O_3$ production rate. However, ground-based measurements are

often limited in time period and spatial extent. The OBM analysis requires measuring nitric oxide (NO) at sub-ppb levels and more than 50 different types of VOCs and with high accuracy, which is difficult to achieve (Wang et al., 2017a).

Satellite remote sensing provides an alternative way to investigate long time periods of $O_3$ formation sensitivity on large spatial scales. For over two decades, satellite-based spectrometers have provided continuous global observations on a daily basis for two species indicative of $O_3$ precursors, $NO_2$ for $NO_x$ (Martin et al., 2004; Lamsal et al., 2014)  and formaldehyde



(HCHO) for VOC (Palmer et al., 2003; Fu et al., 2007). $NO_x$ can be approximated from satellite observation of $NO_2$ column because of the short lifetime of $NO_x$ and high ratio of $NO_2/NO_x$ in the boundary layer (Duncan et al., 2010; Jin and Holloway, 2015). HCHO is an intermediate of the oxidation reaction of various VOCs in the atmosphere. The production of HCHO is approximately proportional to the summed rate of reactions of VOC with peroxy radicals (Sillman, 1995). Therefore, HCHO can be used as a tracer for VOCs in the absence of other VOC observations (Martin et al., 2004; Duncan et al., 2010). The $O_3$

formation sensitivity is defined by the ratio of HCHO to $NO_2$ (referred to as FNR) (Martin et al., 2004). Duncan et al. (2010) combined models and OMI HCHO and $NO_2$ data to show certain ranges of FNR that can be useful for classifying a region into VOC-limited or $NO_x$-limited regime. A FNR smaller than 1 indicates the VOC-limited conditions, and a FNR higher than 2 to indicate the $NO_x$-limited conditions. A FNR in the range of 1 - 2 should generally be considered as indicative of the transitional regime. These FNR thresholds defined by Duncan et al. (2010) have been widely used for various regions (Choi and Souri,

2015; Jin and Holloway, 2015; Souri et al., 2017; Jeon et al., 2018) and with different satellite instruments (Choi et al., 2012).

However, these prior studies linked FNR with surface $O_3$ sensitivity in models (Martin et al., 2004; Duncan et al., 2010). Modeled and observed HCHO columns, $NO_2$ columns and surface $O_3$ often disagree. Jin et al. (2017) found that the spatial and temporal correlations between the modeled and satellite-derived FNR vary over the used satellite instruments. Brown-Steiner et al. (2015) found persistent $O_3$ biases under all configurations of the Global Climate-Chemistry Model (GCCM) with

detailed tropospheric chemistry. Although FNR thresholds defined by Duncan et al. (2010) have been used previously to investigate $O_3$-$NO_x$-VOC sensitivity in China (Witte et al., 2011; Tang et al., 2012; Jin and Holloway, 2015), their conclusions were based on the atmospheric situations in the United States and may not be suitable for the more complicated air pollution in China, concerning the different emission factors, sources, pollution levels and climatology. For example, compared with United States, most cities in China have higher aerosol levels (van Donkelaar et al., 2010; Li et al., 2019c). Secondary aerosol

production may become a large sink of radicals, which could shift $O_3$ production toward a VOC-limited regime under these FNR thresholds suited to United States (Liu et al., 2012; Li et al., 2019a). It is therefore useful to describe surface $O_3$ sensitivity using FNR thresholds derived entirely from satellite observed FNR and ground-based measurements of $O_3$. In addition, Schroeder et al. (2017) using airborne measurements suggested that the range and span of FNR marking the transitional regime varies regionally.

In this study, we assess if space-based $HCHO/NO_2$ ratio captures the non-linearity of $O_3$ chemistry by matching satellite observations with ground-based $O_3$ measurements over the major Chinese cities. Thresholds suited for China between space-based $HCHO/NO_2$ and the ground-based $O_3$ response patterns are derived from observations instead of model results. We focus on the spatial and temporal variability of $O_3$ formation sensitivity using our FNR thresholds on a nationwide scale and in typical cities from 2016 to 2019.

More recently a new unique situation has occurred with the outbreak of the COVID-19 pandemic, which provided a unique opportunity to demonstrate our predicted effects on $O_3$ pollution in China. Efforts to halt the spread of COVID-19 have drastically reduced human activities worldwide (Siciliano et al., 2020; Tian et al., 2020a). As a result of these restrictions, a significant reduction in primary air pollutant emissions, especially in the concentration of $NO_2$, has been noticed in China and





several European and American countries (Tobás et al., 2020; Wang and Su, 2020; Bauwens et al., 2020; Ding et al., 2020).
By contrast, increasing $O_3$ concentrations during the same period were observed in densely metropolitan areas throughout the
world (Siciliano et al., 2020; Zoran et al., 2020; Huang et al., 2020).

Section 2 describes the data and methods used in this study. Section 3 presents our derived FNR thresholds method and
variations of $O_3$ formation sensitivity in China. In addition, impacts of the COVID-19 outbreak on $O_3$ levels are discussed.
Finally, section 4 gives a brief summary.

## 2 Data

### 2.1 Satellite data

We use the $NO_2$ and HCHO observations from the Ozone Monitoring Instrument (OMI) aboard the National Aeronautics
and Space Administration (NASA) satellite AURA, which was launched in July 2004 (Levelt et al., 2006). In an ascending
sun-synchronous polar orbit, OMI passes the equator at about 13:40 local time (LT), providing global measurements of aerosol
parameters, cloud, and various trace gases ($NO_2$ and HCHO among them) (Levelt et al., 2006). The high spatial resolution (13
km $\times$ 24 km) allows for observing fine details of atmospheric parameters (Jin and Holloway, 2015). OMI data are considered
to be reliable and of good quality for the full mission thus far (Zara et al., 2018). In addition, the OMI overpass time is well
suited to detect the $O_3$ formation sensitivity during the afternoon, when $O_3$ photochemical production peaks and when the
boundary layer is high and the solar zenith angle is small, maximizing instrument sensitivity to HCHO and $NO_2$ in the lower
troposphere (Jin et al., 2017).

We use the OMI tropospheric $NO_2$ and HCHO data products from the European Quality Assurance for Essential Climate
Variables project (QA4ECV, http://www.qa4ecv.eu/). $NO_2$ data are compiled by the Royal Netherlands Meteorological
Institute (KNMI). The tropospheric $NO_2$ column density, is defined as the vertically integrated number of $NO_2$ molecules
between the Earth's surface and the tropopause per unit area. We select QA4ECV $NO_2$ daily observations with: (1) no
processing error; (2) less than 10% snow or ice coverage; (3) solar zenith angle less than 80°; (4) cloud radiance fraction less
than 50%. The QA4ECV $NO_2$ monthly datasets are processed with a spatial resolution of 0.125 °$\times$ 0.125 °. Boersma et al.
(2018) reported the single-pixel uncertainties for the QA4ECV $NO_2$ columns are 35% - 45% in the polluted regions, the
monthly mean $NO_2$ columns are estimated to have an uncertainty of ±10%.

The OMI tropospheric HCHO are retrieved by the Belgian Institute for Space Aeronomy (BIRA-IASB) (Smedt et al.,
2017). We select processing_quality_flags = 0 or > 255 providing a selection of observations that is considered as optimal.
Zara et al. (2018) found that the QA4ECV HCHO slant column densities (SCDs) have uncertainties of 8 - 12 $\times 10^{15}$
molecule/$cm^2$ and a remarkably stable trend (increase < 1% $year^{-1}$). The QA4ECV HCHO monthly datasets are available with
a spatial resolution of 0.05 °$\times$ 0.05 °. Temporal averaging has been shown to reduce the HCHO measurements uncertainty and
noise (Millet et al., 2008). We re-grid the monthly OMI HCHO data (0.05 °$\times$ 0.05 °) to the same grid as for the monthly OMI
$NO_2$ data (0.125 °$\times$ 0.125 °).



## 2.2 NOₓ emission

Emission inventories of air pollutants are important sources of information for policy makers and form essential input for air quality models. Bottom-up inventories are usually compiled from statistics on emitting activities and their typical emission factors, but are sporadically updated (Li et al., 2017). Satellite-derived emission inventories have important advantages over bottom-up emission inventories: they are spatially consistent, have high temporal resolution, and provide up-to-date emission information (Mijling and van der A, 2012). In this study, we use monthly mean $NO_x$ surface emission estimates derived from OMI observations of tropospheric $NO_2$ columns by the Daily Emission Estimation Constrained by Satellite Observations (DECSO) algorithm. Mijling and van der A (2012) for the first time developed DECSO (version 1) by calculating the sensitivity of concentration to emission based on a chemical transport model and using trajectory analysis to account for transport away from the source. Ding et al. (2015) improved DECSO (version 3) and demonstrated that it is able to detect the monthly change of $NO_x$ emissions due to air quality regulations on a city level. The $NO_x$ emissions derived by the improved DECSO version 5 are in good agreement with other bottom-up anthropogenic emission inventories. In addition, the improved algorithm is able to better capture the seasonality of $NO_x$ emissions. The precision of monthly $NO_x$ emissions derived by DECSO version 5 for each grid cell is about 20 % (Ding et al., 2017). Here, we use $NO_x$ emissions derived by the latest DECSO version 5.1qa which provides monthly emissions for the last decade (2007-2020) (Ding et al., 2018). These datasets are available from http://www.globemission.eu/region_asia/datapage.php.

## 2.3 Ground-based observations

Since 2012 the Chinese government at various levels began to establish a national air quality monitoring network, which released real-time ground-level $O_3$ monitoring data to the public. By 2016, the establishment of more than 1,000 sites have been completed, covering more than 300 cities across the country.

We use hourly $O_3$ and $NO_2$ concentrations (in standard condition, 273 K, 101.325 kPa) from the network of ~1000 sites operated by the China Ministry of Ecology and Environment (CMEE) since 2016. CMEE revised the monitoring of pollutants to a new reference condition (298K, 101.325 kPa) since 1 September 2018 (CMEE, 2018). Daily ground-based $O_3$ and $NO_2$ observations are calculated from hourly observations at OMI overpass time (average of 13:00 LT and 14:00 LT). In this study, we convert the gas concentrations before 1 September 2018 from the standard condition to the reference condition. The temperature dependence is according to Charles's law (1)

$$\frac{V_{std}}{T_{std}} = \frac{V_{ref}}{T_{ref}} \quad (1)$$

where $V_{std}$ is the volume of a gas under standard condition, $V_{ref}$ is the volume of a gas under reference condition, $T_{std}$ (unit: K) is the thermodynamic temperature of standard condition, $T_{ref}$ (unit: K) is the thermodynamic temperature of reference condition. The gas concentration conversion follows

$$\frac{c_{std}}{c_{ref}} = \frac{M/V_{std}}{M/V_{ref}} = \frac{V_{ref}}{V_{std}} \quad (2)$$





where $C_{std}$ is the gas concentration under standard condition, $C_{ref}$ is the gas concentration under reference condition.

Because the Chinese national air quality monitoring network stations are mostly located in the center of cities or densely populated areas, usually the most polluted regions, we select the NaHa station located on the small island Okinawa in Japan, as a location with a clean atmosphere. The hourly $O_3$ and $NO_2$ observations of NaHa station are provided by the Japanese Atmospheric Environmental Regional Observation System (AEROS, http://soramame.taiki.go.jp/Index.php).

## 2.4 CLASS model

We simulate the nonlinear relationship among $O_3$, $NO_2$ and HCHO using the Chemistry Land-surface Atmosphere Soil Slab model (CLASS). We performed a series of numerical experiments with the same dynamic and chemistry conditions listed in Table 1, but modified only the concentrations of $NO_2$ and HCHO.

The CLASS model solves the diurnal evolution of dynamical variables (temperature, specific humidity and wind) and chemical species over time in a well-mixed, convective Atmospheric Boundary Layer (ABL) in which entrainment and boundary layer growth are considered (Vilà-Guerau de Arellano et al., 2015; van Heerwaarden et al., 2010). All these variables are assumed to be constant with height due to intense turbulent mixing driven by convection (van Heerwaarden et al., 2010). The surface is assumed to be homogeneous in this box model. Chemistry is represented by a chemical scheme based on 27 reactions that control $O_3$ formation described by van Stratum et al. (2012), with $O_3$, NO and $NO_2$ as most important species. This simplified chemical scheme is able to represent the evolution of chemical species in semirural areas (Janssen et al., 2012; van Stratum et al., 2012). The model has been validated under various dynamical conditions (Barbaro et al., 2014; Janssen et al., 2012; van Heerwaarden et al., 2010).

Table 1. Configuration and settings of CLASS modeling system

| Item | Status/Values |
| --- | --- |
| Total simulation time | 12h |
| Time step | 60s |
| Initial ABL height | 200m |
| Mixed-layer | On |
| Initial mixed-layer potential temperature | 288K |
| Initial temperature jump at height | 1K |
| Wind | Off |
| Surface scheme (sea or land) | Off |
| Chemistry | On |



## 3 Results

### 3.1 $O_3$ formation sensitivity regime classification

In Figure 1a, the CLASS model is applied to generate $O_3$ isopleths, which illustrate $O_3$ as function of $NO_2$ and HCHO values. The isopleths show that $O_3$ formation is a highly nonlinear process in relation to $NO_2$ and HCHO. When $NO_2$ is low, the $O_3$ increases with increasing $NO_2$. As $NO_2$ increases, the $O_3$ eventually reaches a local maximum. At higher $NO_2$ concentrations, the $O_3$ would decrease with increasing $NO_2$.

We first evaluate if satellite-based HCHO and $NO_2$ columns can capture the nonlinear $O_3$-$NO_2$-HCHO chemistry shown by the CLASS model. In order to obtain a representative observation sample, we create monthly mean ground-based $O_3$ and $NO_2$ observations of 360 cities across China from the Chinese national air quality monitoring network from 2016 to 2019, and the background station observations from NaHa, Japan for comparison. Temperature is also a major factor in $O_3$ chemistry. $O_3$ pollution is rare when the ambient temperature is below 20 ℃ (Sillman, 2003). The seasonality of ground-level $O_3$ concentrations also exhibited monthly variability peaking in summer and reaching the lowest levels in winter over China (Wang et al., 2017b). In addition, long $NO_x$ lifetime and low concentrations of OH and $RO_2$ radicals would lead most regions of China to a VOC-limited regime in winter (Shah et al., 2020). Therefore, we focus in this study on May - October as the summer period when meteorology is favorable for $O_3$ formation (Jin et al., 2017).

By directly connecting HCHO columns from OMI observations with ground-based measurements of $NO_2$ and $O_3$ from 360 cities across China during May – October from 2016 to 2019 in Figure 1b, we find that the satellite-based HCHO columns and ground-based $NO_2$ concentrations can capture nonlinear $O_3$ chemistry consistent with the CLASS model results. It indicates that tropospheric HCHO columns from OMI can represent the near-surface HCHO environment as revealed by previous studies (Martin et al., 2004; Duncan et al., 2010; Jin et al., 2017). The overall $O_3$-$NO_2$-HCHO chemistry is also captured by satellite-based HCHO and $NO_2$ columns in Figure 1c, which reflects the reliability of $NO_2$ satellite retrievals.

Having established this relationship between satellite-based HCHO/$NO_2$ columns and surface $O_3$ concentrations, we subsequently derive the FNR thresholds marking the $O_3$ transitional regime. The local $O_3$ maximum can be thought of as a dividing line separating two different photochemical regimes (Sillman, 1999). According to the National Ambient Air Quality Standards released in 2012, 1-hour average $O_3$ concentration should below 160 μg/$m^3$ in rural regions and below 200 μg/$m^3$ in urban regions (Li et al., 2018). We assume that the monthly $O_3$ (daily $O_3$ data is averaged at 13:00 LT and 14:00LT) exceeding 160 μg/$m^3$ has a large component that is due to local photochemical production, not meteorology or regional transport. We investigate the maximum, top 5%, top 10% and top 15% of the monthly $O_3$ with corresponding FNRs for each city during May - October from 2016 to 2019 in Figure S1 in the supplement. The FNR calculation for each city is restricted to pixels where monthly HCHO columns are higher than $2 \times 10^{15}$ molecule/$cm^2$ (detection limitation) and $NO_2$ columns are more than $1.5 \times 10^{15}$ molecule/$cm^2$ (which are defined as polluted regions). We find that the top 10% dataset contains more than half of the total monthly high $O_3$ concentrations (> 160 μg/$m^3$) data and more than 80% of the data in the top 15%. Therefore, we will define the transitional regime based on the monthly $O_3$ exceeding 160 μg/$m^3$ in the top 10% dataset in Figure 1d.





It should be noted that the actual split between $NO_x$-limited and VOC-limited regime includes a broad transitional region rather than a sharp dividing line (Sillman, 1999). Although we reduce the noise by gridding, there is a blurry transition between $NO_x$-limited and VOC-limited regimes. The lack of sharp and clear transitions between two $O_3$ sensitivity regimes is likely influenced by factors such as meteorology, chemical and depositional loss of $O_3$ and noisy satellite data. Taking into account the large range of transitional regime and universality of derived FNR thresholds in China, the FNR thresholds [2.3, 4.2]

marking the transitional regime, are defined as the $\pm 30\%$ range from the median (3.28) covering more than half of transitional regime FNRs. Three regimes can be roughly identified from the FNR thresholds we adopted: a VOC-limited regime should occur when the FNR < 2.3 and a $NO_x$-limited regime should occur when the FNR > 4.2. The FNR between 2.3 and 4.2 reflects the transitions between the two regimes.

### 3.2 Variations of $O_3$ formation sensitivity in China

Figure 2a and 2b show the photochemical regime classification over China in summer of 2016 and 2019 using our FNR thresholds. Combined with the China provincial administrative division in Figure S2 in the supplement, we see the VOC-limited regimes mainly appear in the North China Plain (NCP), the Yangtze River Delta (YRD) and the Pearl River Delta (PRD), and the $NO_x$-limited regimes dominate the remaining areas, which are consistent with results from Wang et al. (2019a) and Jin and Holloway (2015). In the NCP, the VOC-limited regimes are found in Beijing and some big cities in Hebei province,

central regions in Shandong province and Henan province. Transitional regimes control the remaining regions of Shandong province and Henan province and most regions of Hefei province. In the YRD, the VOC-limited regimes are found in Shanghai and southern Jiangsu province. In the PRD, the VOC-limited regimes are found in Guangzhou. Outside the NCP, YRD and PRD, the VOC-limited regimes concentrate in city centers of Shenyang, Chengdu, Chongqing, Xi'an and Wuhan, that are surrounded by transitional regimes in the suburban areas. It has been acknowledged that the urban $O_3$ formations are generally

VOC-limited due to the large amount of $NO_x$ emissions from diverse sectors, like transportation, industry, residential sector and power plants (Shao et al., 2009; Wang et al., 2009; Sun et al., 2011). The $NO_x$-limited or transitional regimes dominated $O_3$ formation in the suburban and rural areas of eastern China (Xing et al., 2011; Jin et al., 2017).

Comparison of $O_3$ sensitivities between 2016 and 2019 shows noticeable changes from VOC-limited regime to transitional regime in the NCP, YRD and PRD. In the NCP, the continuous area of VOC-limited regimes that occurred in 2016

change to transitional regimes in 2019. The VOC-limited regimes remain in central Beijing, Tianjin, Shijiazhuang, Jinan and Zhengzhou. In the YRD, Shanghai and Nanjing remain in the VOC-limited regime, other cities mostly change to the transitional regime. In the PRD, the VOC-limited regime still controls Guangzhou, while the transitional regimes control its surrounding cities.

Figure 2c and 2d show mean HCHO columns over China in the summer of 2016 and 2019. The columns exceed $15 \times$

$10^{15}$ molecule/cm$^2$ in megacity clusters, such as in the NCP, YRD and PRD, and the Sichuan Basin. Shen et al. (2019) found large increases of HCHO columns during May - September over 2005 - 2016 in the NCP and the YRD, consistent with the trend of anthropogenic VOC emissions. Our results show that the satellite HCHO columns increase in the NCP and the YRD





and decrease in the PRD and in the Sichuan Basin during May - October of the 2016 - 2019 period. Figure 2e shows mean $NO_2$ columns over China in the summer of 2016. The NCP, YRD, PRD, Sichuan Basin and Urumqi have high levels (80 ×

$10^{15}$ molecule/cm$^2$) of $NO_2$ columns. Figure 2f shows the satellite $NO_2$ columns have a strong decline in the NCP, the PRD, Hunan, Hubei and Jiangxi province in summer from 2016 to 2019. However, the YRD shows increasing $NO_2$ columns in 2019.

We select typical cities (Beijing, Shanghai, Guangzhou, Neijiang, Lhasa and NaHa) to analyze in more detail the $O_3$ formation sensitivity in the summers of 2016 to 2019 in Figure 3. These cities are selected based on their different chemical regimes in 2016. The locations of the six cities are shown in Figure S3 in the supplement. Economically developed megacities

or provincial capital cities such as Beijing, Shanghai and Guangzhou with high levels of tropospheric $NO_2$ and HCHO, remain in the VOC-limited regime over 2016-2019. The reduction of tropospheric $NO_2$ results in a shift in the $O_3$ formation sensitivity in cities such as Neijiang over 2016-2019. Lhasa as a city with low $NO_2$ and the background station in NaHa with even lower HCHO and $NO_2$ columns remain in the $NO_x$-limited regime over 2016-2019.

As we know, $O_3$ increases with increasing $NO_x$ in the $NO_x$-limited regime and decreases with increasing $NO_x$ in the VOC-

limited regime. The contrast between $NO_x$-limited and VOC-limited regimes illustrates the difficulties involved in developing policies to reduce $O_3$ in $NO_x$ polluted regions. Reductions in VOCs will only be effective in reducing $O_3$ if VOC-limited chemistry predominates. Reductions in $NO_x$ will be effective only if $NO_x$-limited chemistry predominates and may actually increase $O_3$ in VOC-sensitive regions. If cities belonging to the VOC-limited regime like Beijing only focus on the reduction of $NO_x$ while ignore the control of VOC emissions, they will experience a process of rising $O_3$ concentrations, the more $NO_x$

decrease, the greater the increase of $O_3$ will be.

### 3.3 Observed response of ground-level $O_3$ to chemical formation sensitivity

To validate the regimes derived from satellite observations, we also analyze the surface $NO_2$ observations from ground-based measurements. Figure 4a and 4b show the mean ground-based $NO_2$ concentrations in summer of 2016 and 2019. According to the $NO_x$ surface emission estimates derived with DECSO from OMI observations, the $NO_x$ emissions in eastern

China (18 °N, 104 °E, 41.5 °N, 124 °E) decrease from 5.93 Tg/yr in 2016 to 4.21 Tg/yr in 2019. Such a strong decline in $NO_x$ emissions led to decreasing ambient $NO_2$ concentrations at NCP (Beijing, Shijiazhuang, Zhengzhou, Jinan) and YRD (Hefei and other cities in Anhui province). In Figure 4c, the national average $NO_2$ concentration decrease by 14.4% in summer from 2016 to 2019.

Figure 4d and 4e show the mean ground-based $O_3$ concentration of about 360 cities across China in summer of 2016 and

2019. Generally, the $O_3$ levels in western China are lower than in eastern China. In 2016, few cities have an average $O_3$ concentration above 140 μg/m$^3$. In 2019, cities with a mean $O_3$ concentration exceeding 140 μg/m$^3$ occurred at the NCP (Tianjin, Shijiazhuang, some cities in Shandong and Henan province), the YRD (Nanjing), and the PRD (Guangzhou). In Figure 4f, we see the number of cities with average $O_3$ values above 140 μg/m$^3$ increases rapidly from 2.20% in 2016 to 31.37% in 2019. The cities with an average $O_3$ value below 80 μg/m$^3$ decrease from 11.02% in 2016 to 2.24% in 2019. In addition, the

nationwide $O_3$ average in summer increases year by year from 2016 (104.86 μg/m$^3$) to 2019 (125.14 μg/m$^3$). Li et al. (2019a)





reported the increasing $O_3$ trends in summer in megacity clusters of eastern China and the highest $O_3$ concentrations are in the NCP, which are consistent with our results.

A complex coupling of primary emissions, chemical transformation, and dynamic transport at different scales determine the $O_3$ pollution (Jacob, 1999). $NO_x$ and VOCs play important roles in $O_3$ formation. Emissions of $NO_x$ and VOCs to the
environment are the starting point of $O_3$ pollution problems. During the past decade in China, ambitious steps have been taken to control $NO_x$ emissions. In 2013, the Chinese State Council issued the APPAP. Stringent control measures were carried out since then, including phasing out high-emitting industries, closing outdated factories, tightening industrial emission standard, improving fuel quality (Wang et al., 2019a). However, to the other important $O_3$ precursors, VOCs, less attention has been given in emission control strategy. Li et al. (2019b) concluded that anthropogenic NMVOC emissions in China during 1990-
2017 have been increasing continuously due to the dramatic growth in activity rates and absence of effective control measures. Following China's past control strategy on VOCs, we can regard VOC emissions as rising or in steady state.

The reduction of the $NO_x$ emissions for cities in the VOC-limited regime is one of the main reason for the increasing of $O_3$. Figure 5a shows the difference of total $NO_x$ emissions derived from OMI observations in summer in east China between 2019 and 2016. A decline in $NO_x$ emissions centers at the NCP, YRD and PRD, where most areas belong to the VOC-limited
regime. In order to provide further insight into the impact of $NO_x$ emission variations on $O_3$ concentrations, five selected typical cities (Beijing, Shanghai, Guangzhou, Neijiang and NaHa) are shown in more detail (see Figure 5b and 5c). For cities under the control of VOC-limited chemistry (Beijing, Shanghai and Guangzhou), accompanied with decreasing $NO_x$ emissions, $O_3$ concentrations generally show an opposite behavior to $NO_x$ emissions. The $O_3$ formation sensitivity in Neijiang shows a shift from the transitional to the $NO_x$-limited regime over 2016-2019. The reduction of $NO_x$ emissions in the transitional regime
is accompanied by decreasing $O_3$ in Neijiang. Although the $O_3$ data in NaHa for 2016-2018 are unavailable, we see that $O_3$ concentrations in NaHa are low in 2019 and $NO_x$ emissions are stable during 2016-2019.

**3.4 Enhanced $O_3$ levels during COVID-19 lockdown in China**

The measures in response to the outbreak of the COVID-19 lead to sudden changes of $NO_x$ emissions and anthropogenic HCHO emissions in China in the beginning of 2020 (Wang et al., 2020; Hui et al., 2020). We analyze the change of $O_3$
concentrations during the lockdown period to validate our method. To look into COVID-19 lockdown impacts on short-term $O_3$ level, we choose two time periods covering 357 cities across China: period I (3 - 23 January, 2020) and period II (9 - 29 February, 2020), to avoid the coincidence of Chinese New Year holidays (24 January to 8 February, 2020).

Figure 6a shows enhanced $O_3$ levels in most cities of eastern China during the COVID-19 lockdown, except for some cities in PRD and Fujian province. The cities with $O_3$ concentration increases of more than 40 μg/m³ are located in the NCP
and the YRD, the populous regions of China, indicating a potential negative health effect from $O_3$ exposure in these regions. Figure 6b shows strong reductions in $NO_x$ emissions in eastern China, especially in Henan, Hubei and Jiangsu province, where as a consequence of the lockdown, transportation, construction, and light industry activities have been dramatically decreased.





Using our observation-based FNR thresholds method, we see that most regions of eastern China belong to the VOC-limited regime during period I and II in Figure 6c and 6d. Previous studies also reported that the $O_3$ chemistry in the urban

areas in China in wintertime is in a VOC-limited regime due to the relative lack of $HO_x$ radicals (Seinfeld and Pandis, 2016). During winter (VOC-limited conditions), when the concentration of $NO_x$ is high, and the level of UV radiation is low, the $O_3$ production varies inversely with the $NO_x$ concentration (Sillman et al., 1990). The $NO_x$ reduction during the lockdown is higher than the VOC reduction (Sicard et al., 2020). Thus, a reduction in $NO_x$ leads to an increase of the $O_3$ concentrations in most regions of eastern China during period II. Besides, reduction of freshly emitted NO in particular from road traffic

alleviates $O_3$ titration locally (Seinfeld and Pandis, 2016; Levy et al., 2014). The $O_3$ titration occurs particularly in winter (less photolysis reactions of $NO_2$) under high $NO_x$ levels (Sillman, 1999). However, the lockdown measures result primarily in a lower titration of $O_3$ by NO due to the reduction in local $NO_x$ emissions by road transport, which also enhances $O_3$ levels in urban areas. On the other hand, some cities, mainly located in southeastern China, showed decreasing $O_3$ levels. Zhao et al. (2020) concluded that the cause of $O_3$ decline in these cities is the emission changes of $NO_x$ and VOC. In Figure 6c we see

that some cities in Fujian and Guangdong provinces belong to the transitional regime. Theoretically, the transitional regime should correspond to the conditions at which $O_3$ formation is most efficient, indicating that reductions or increases in $NO_x$ and VOCs will reduce the $O_3$ concentration.

## 4 Conclusion

Satellite-based $HCHO/NO_2$ ratios and ground-based $O_3$ measurements were directly connected to capture the non-

linearity of surface $O_3$ chemistry over major Chinese cities in this study. Evaluating the FNR thresholds marking the $O_3$ transitional regime in which $O_3$ formation is less sensitive to the precursors, we found a broad transitional region, which reflects differences of factors among 360 cities, such as emissions, meteorology, and regional transport. The national FNR thresholds are defined as follows: a VOC-limited regime should occur for FNR < 2.3 and a $NO_x$-limited regime should occur for FNR > 4.2. The FNR between 2.3 and 4.2 reflects the transition between the two regimes. Our FNR thresholds derived from satellite

and ground-based observations are higher than previous reported model-based values. The nonlinear chemistry of $O_3$ depends on its precursors $NO_2$ and VOCs with contributions from both local and regional sources (Xue et al., 2014). Modeling studies are good at simulating the response of surface $O_3$ to an overall reduction in $NO_x$ or VOC emissions. The FNR thresholds derived with *in situ* $O_3$ observations will be more indicative of the local $O_3$ chemistry than the model, including the effect of $NO_x$ titration over urban areas (Jin et al., 2020).

We analyzed the spatial and temporal variability of $O_3$ formation sensitivity using our FNR thresholds over China from 2016 to 2019. Our results showed that $O_3$ formation sensitivity tends to be VOC-limited over urban areas and $NO_x$-limited over rural and remote areas in China. In 2016, the VOC-limited regimes mainly appear in the NCP, the YRD and the PRD. In 2019, there was a shift in most NCP regions from the VOC-limited to the transitional regime. The area with a VOC-limited regime in the YRD and PRD also shrank. We found that $O_3$ formation sensitivity changes in these regions were associated



with a strong decline in tropospheric $NO_2$ columns in the NCP and the PRD. For megacities such as Beijing and Guangzhou, although they remained in VOC-limited regime over 2016-2019, there was still a decrease in $NO_2$ columns. Consistent with decreasing tropospheric $NO_2$ columns, the national average surface $NO_2$ concentration decreased by 14.4% in summer from 2016 to 2019 and the $NO_x$ emissions in eastern China decreased from 5.93 Tg/yr in 2016 to 4.21 Tg/yr in 2019. This detected spatial expansion of the transitional regime and $NO_x$ emission reduction in VOC-limited regime has contributed to rising

surface $O_3$ concentrations. The nationwide averaged $O_3$ concentration in summer increased year by year from 2016 (104.86 $\mu g/m^3$) to 2019 (125.14 $\mu g/m^3$). The cities with average $O_3$ values above 140 $\mu g/m^3$ increased rapidly from 2.20% in 2016 to 31.37% in 2019.

Emission sources of HCHO, as a tracer of VOCs, can be anthropogenic and biogenic. Shen et al. (2019) found that the OMI HCHO distribution follows their anthropogenic inventory in megacity clusters over China, while it does not follow the

biogenic emissions inventory. Despite the fact that local sources of anthropogenic VOCs are difficult to identify, our FNR thresholds derived from satellite-based information have the potential to provide important information to air quality planners. Compared with stringent control measures for $NO_x$ emissions, VOC emissions got less attention as the other $O_3$ precursor in China. The case study of $O_3$ level changes during the COVID-19 lockdown in China demonstrated that the reductions in anthropogenic $NO_x$ emissions resulted in significant $O_3$ enhancement in urban areas. It indicates that a protocol with strict

measures to control $NO_x$ emissions, without simultaneous VOC emissions controls for power plants and heavy industry, such as petrochemical facilities, achieves only limited effects on $O_3$ pollution.

**Data availability**

Satellite data used in this research can be obtained from public sources. The OMI tropospheric $NO_2$ product from the QA4ECV project can be obtained from http://www.qa4ecv.eu/ecv/no2-pre/data and the HCHO product from

http://www.qa4ecv.eu/ecv/hcho-p/data.

The monthly mean $NO_x$ emission products derived from OMI observations by DECSO v5.1qa can be obtained from http://www.globemission.eu/region_asia/datapage.php.

The hourly $O_3$ and $NO_2$ observations of Naha station are provided by the Japanese Atmospheric Environmental Regional Observation System (AEROS, http://soramame.taiki.go.jp/Index.php).

**Author contributions**

WW and RA provided satellite data, tools, and analysis. RA, JD, MW and TC undertook the conceptualization and investigation. WW prepared original draft. RA and JD carried out review and editing. All authors discussed the results and commented on the paper.



**Competing interests**

The authors declare that they have no conflict of interest.

**Acknowledgements**

The support provided by China Scholarship Council (CSC) during a visit of Wannan Wang to Royal Netherlands Meteorological Institute (KNMI) is acknowledged.

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





**Figure 1: (a) The simulated O₃ isopleths versus NO₂ and HCHO using the CLASS model. (b) The 360 cities' monthly mean *in-situ* O₃ concentrations versus *in-situ* NO₂ concentrations and HCHO columns from OMI observations in the summer during 2016-2019.**
620 **Note: daily ground-based O₃ and NO₂ observations are calculated from hourly observations at OMI overpass time (averaged at 13:00 LT and 14:00 LT). (c) same as (b), but with NO₂ columns from OMI observations. (d) The top 10% monthly O₃ values and corresponding FNRs of each city.**



**Figure 2: (a) Photochemical regime classification over China in the summer of 2016. (b) Same as (a), but for 2019. Note: no data**
**grids in (a) and (b) corresponds to monthly HCHO columns below the detection limit (2 × 10 $^{15}$ molecule/cm$^2$) or NO$_2$ columns lower**
**than 1.5 × 10 $^{15}$ molecule/cm$^2$. (c) Mean HCHO columns over China in the summer of 2016. (d) Same as (c), but for 2019. (e) Mean**
**NO$_2$ columns over China in the summer of 2016. (f) Same as (e), but for 2019.**





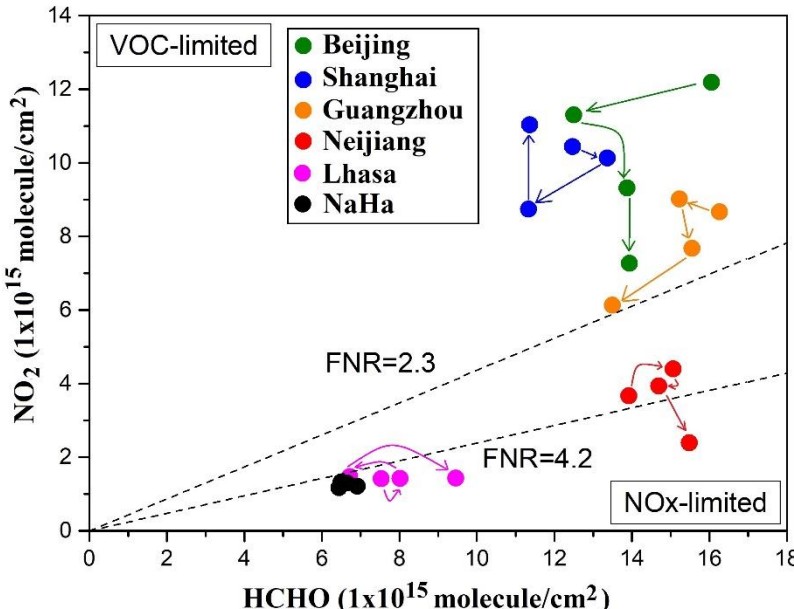

**Figure 3: The change of O₃ formation sensitivity of six cities (Beijing, Shanghai, Guangzhou, Neijiang, Lhasa and NaHa) in summer**
630 **from 2016 to 2019.**

**Figure 4: (a) Mean ground-based NO₂ concentration at each city in the summer of 2016. (b) Same as (a) but for 2019. (c) The bars indicate the number of cities (left axis) in a certain NO₂ range in summer from 2016 to 2019. The black line indicates the average NO₂ concentration (right axis) of all cities. (d) Mean ground-based O₃ concentration at each city in summer of 2016. (e) Same as (d) but for 2019. (f) Same as (c) but for O₃. Note: daily *in-situ* NO₂ and O₃ data is the average of 13:00-14:00 of the sites in each city.**


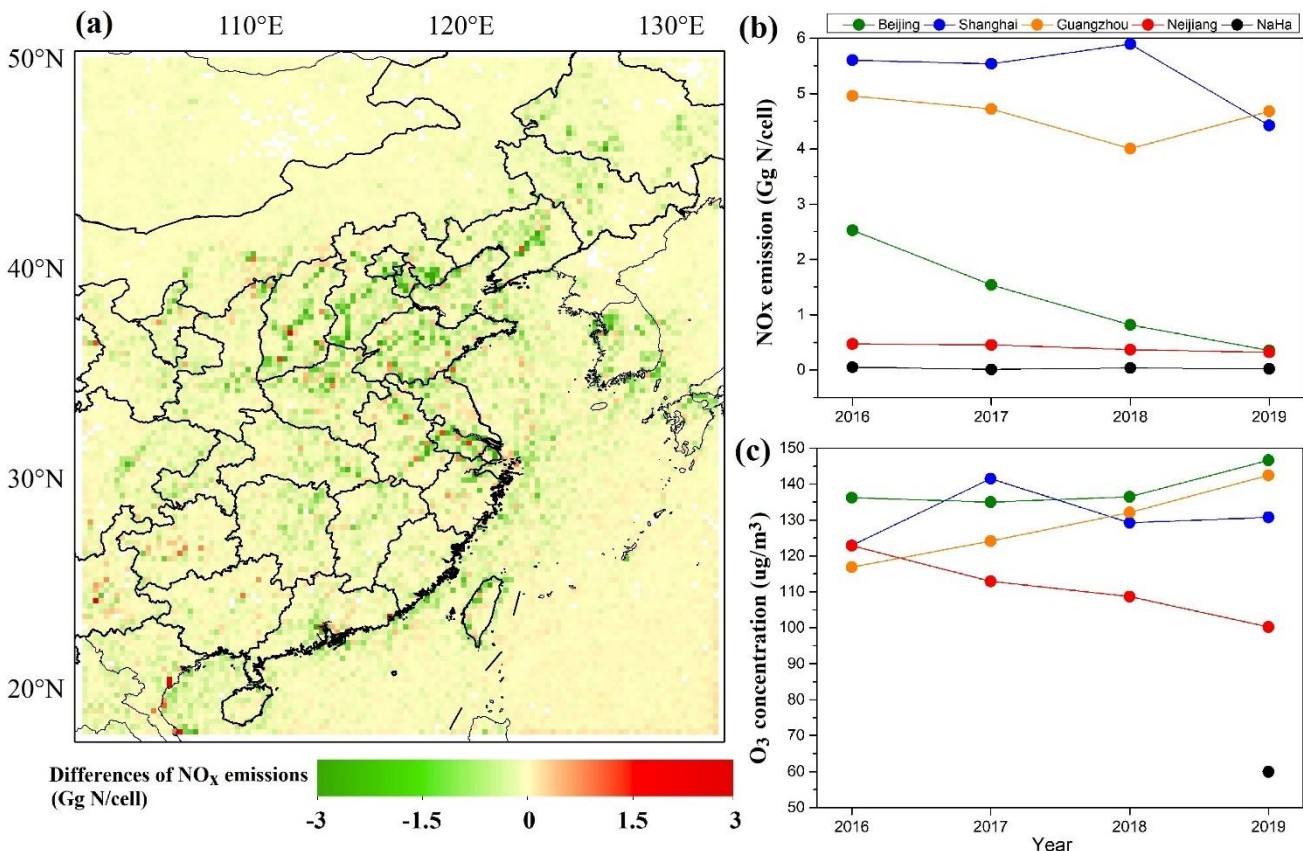

**Figure 5: (a) Differences of total NO$_x$ emissions derived from OMI observations in summer in east China between 2019 and 2016. (b) Variations of total NO$_x$ emissions in five cities (Beijing, Shanghai, Guangzhou, Neijiang and NaHa) in summer from 2016 to 2019. (c) Variations of mean ground-based O$_3$ concentrations in five cities in summer from 2016 to 2019.**







**Figure 6: (a)** Differences of mean ground-based O₃ concentrations in east China between period I and period II. **(b)** Differences of mean NOₓ emissions in east China between period I and period II. **(c)** O₃ formation sensitivity in east China during period I. **(d)** Same as (c), but for period II.