# Peer review of "Figure S1. (a) The maximum of monthly O3 values and corresponding FNRs of each city. (b) same as (a), but for top 5%. (c) same as (a), but for top 10%. (d) same as (a), but for top 15%."

_Atmospheric Chemistry and Physics, 2020_

## Referee Comment (RC1) · Anonymous Referee #1 · 18 Jan 2021

In their manuscript "Spatial and temporal changes of the ozone sensitivity in China based on satellite and ground-based observations", Wannan Wang et al. report on a study using OMI $NO_2$ and HCHO columns to investigate where in China photochemical ozone production is $NO_x$ limited, and where it is VOC limited. In contrast to earlier studies using similar approaches, the classification here is based on linking satellite columns with in-situ surface ozone observations to empirically determine thresholds for the HCHO to $NO_2$ ratio. The method is applied to data from China for the years 2016 to 2019 to investigate changes in photochemical ozone regime. Measurements from the lockdown phase in spring 2020 are included as a test case on the effect of strongly reduced $NO_x$ emissions.

[Figure]

The manuscript is clearly written and fits well into the scope of ACP. The results are not really surprising but add information on a relevant topic in particular in view of what the best approach is to reduce ozone levels in China. I have however concerns with the basic approach taken to identify the NOx and VOC limited regimes, which is the key point of this study and not trivial. In my opinion, this needs to be better explained and justified before this manuscript can be accepted for publication in ACP.

**Major comment**

The one new thing in this study is that the different ozone chemistry regimes are defined from measurements alone and not from models which explicitly determine it for each location. This is an interesting approach but I do not see strong justification for it in the manuscript. What the authors use in Figure 1d is a display of monthly mean ozone surface concentrations as a function of the ratio of formaldehyde to NO2 columns. While it is tempting to look for a maximum in the ozone curve and define this as the separation between NOx and VOC sensitive domains, this is not necessarily justified. The FNR determines the *change* of ozone levels in reaction to a change in NOx or VOC concentrations, but not the total ozone concentration itself (as can clearly be seen in the graph). An argument can be made that by limiting the analysis to the highest values, we are indeed looking at local ozone chemistry and assuming that everything else remains unchanged, absolute ozone levels should reflect ozone production but more justification is needed to make this approach convincing. I think that adding the threshold lines in Figure 1c may help to make the argument.

Thresholds are defined from this figure in a manner not clear to me but this is of course the key question: What are the correct thresholds, are they the same throughout China, are they valid in all seasons / meteorological conditions, do they change over time as emission patterns change? None of this is discussed and this needs to be added.

**Minor comments**

At some point in the manuscript, a short discussion of the problems in using columnar

data instead of surface concentrations is needed, as well as of the question, in how far monthly averages are representative of the highly variable concentrations and the strongly non-linear NOx-O3 chemistry.

L111: Add that quoted resolution for OMI is at nadir

L137: Which product is used in the DESCO algorithm – the QA4ECV product?

L150: Please add some basic information on which type of instrument is used for the ground-based data, which measurement principle is applied and if there are possible cross-sensitivities

L168: If the model is to be used in any quantitative sense, then much more information on initialisation, VOCs used, inclusion of heterogeneous reactions etc. is needed.

L219: Sentence is not clear to me but probably touches on my main point of concern: That the way the thresholds are derived is oversimplified and not necessarily valid for all locations in China

L303: Up to this point, only summer values have been discussed and used. Now, the method is suddenly used for winter values which is problematic and needs at least to be acknowledged and discussed

L358: Here (and earlier), the assumption Is made that VOC emissions have not decreased (much) during the lockdown which would deserve a bit of discussion, in particular as a significant number of studies related to the signature of the lockdown in pollution in China is already available in the literature.

L363: I guess that something should also be said about the availability of the data produced in this study

Figure 1: This figure needs to be revised in several ways:

1. the same colour scheme and scale should be used for model and observations

2. the figures showing observations should not just plot the data on top of each other as this prevents readers from seeing many of them. Instead, a "heat map" type display would be more appropriate showing the binned data

3. I'd suggest to include the threshold ratio lines also in panel 1c

Figure 2: I don't see the need for the inserted maps in particular as they depict disputed areas without relevance for the study.

Figure 2: Add in caption that HCHO columns are from OMI

Figure 3: Explain arrows in caption

Figure 5: Looking at panels (b) and (c) it is clear that the ozone changes in China are not linked in a simple way to the NOx changes: Ozone decreased very much in Neijiang in spite of very moderate NOx decreases while it increased significantly in Guangzhou where NOx emissions remained basically constant. In Beijing, a reduction in NOx emissions by a factor of 2 from 2016 to 2018 had no effect on surface ozone but further reduction then lead to an increase in O3. This needs more discussion in the text.

Figure 6: Define time periods in caption
* * *

---

## Referee Comment (RC2) · Anonymous Referee #2 · 18 Jan 2021

Anonymous Referee #2: Comments to: Wannan Wang et al. "Spacial and temporal changes of the ozone sensitivity in China based on satellite and ground-based observations" MS No.: acp-2020-1097 MS type: Research article Special Issue: Regional assessment of air pollution and climate change over East and Southeast Asia: results from MICS-Asia Phase III

General comments:

The paper provides a substantial contribution to the understanding of photochemical ozone production in China. Instead of looking directly at ozone precursors (NOx and VOCs) in the field, it is demonstrated here that information on the photochemical regime can be derived from satellite data of NO2 and HCHO column densities. In this context, the authors develop a method to discriminate between VOC and NOX dependence using characteristic monthly HCHO/NO2 column density ratios from satellite observations. An important aspect of this approach is that they contrast satellite-based HCHO/NO2 ratios for a large number of ground-based ozone measurements and derive characteristic thresholds for VOC or NOx dependence of ozone formation. In the last part of the paper, the developed methodology is applied to the exceptional situation of the COVID-19 lockdown in late winter 2020 in China. The increase in ozone concentrations observed for numerous areas in eastern China was contrasted with the parallel observed decrease in NO2 column densities, and resulting changes for the pre/post lockdown photochemical regimes prevailing in China were derived from the satellite-based HCHO/NO2 ratios. The authors conclude by pointing out that China's future ozone reduction strategy should in any case be accompanied by a parallel VOC reduction control in addition to the focus on NOx reductions that has prevailed so far.

The paper is well written and structured and clearly identifies the scientific sources used. The authors make a clear distinction between their own contributions and adequately acknowledge previous work from the literature.

Overall, the following ratings are given: Scientific significance: Excellent (4) Scientific quality : Good (3) (see comments) Presentation quality : Excellent (4)

Special comments:

Line 113: The authors point out that only data from the afternoon are used to determine the HCHO/NO2 ratios in order to describe the peak of photochemical ozone production. However, HCHO is a trace gas that is both emitted and photochemically formed in parallel with ozone production. Would the authors please comment on why they do not distinguish between directly emitted HCHO and photochemically formed HCHO. Shouldn't only photochemically formed HCHO be a measure of the intensity of ozone formation? And would it not be possible to draw conclusions about the proportion

of directly emitted HCHO (e.g. from vehicle exhaust gases) from additional "satellite winter data" to be evaluated?

Line 120: The authors point out that they use solar zenith angles of $< 80°$ to determine the monthly HCHO/NO2 column density ratios. However, due to the finite pixel size, the intensity of HCHO production depends on the integral solar radiation lasting for several hours. Therefore, should not different HCHO/NO2 ratios be used due to the enormous extent of the study area ($< 20°$N - $> 45°$N) ?

Line 129: The authors point out that the monthly mean values of the HCHO column densities of ($0.05°$ * $0.05°$) have been converted to the pixel size for NO2 of ($0.125°$ *$0.125°$). Can it be excluded that this procedure leads to significant changes in the HCHO/NO2 ratios (after all, the ratio of the column densities of HCHO/NO2 contains the quotient of the precursor (NO2) and the product of photochemical processing. The adaptation of the HCHO pixel size to that of NO2 could lead to a systematic underestimation of the HCHO column densities. Would the authors please comment on this.

Line 175: Using the CLASS model, the authors demonstrate that photochemical ozone production can be represented in terms of O3 isopleths as a function of HCHO (as a proxi for VOC) and NO2. Shouldn't the isopleths rather represent the ozone production over a period of time instead of the total ozone concentrations shown? Elsewhere, the authors explicitly point out that they distinguish between background ozone and additional ozone production by local photochemical ozone production in the measured ozone monthly means (c. f. line 207). It is suggested to describe the non-linearity of ozone production either only schematically (in the form of a cartoon) or actually quantitatively by naming all boundary conditions (starting concentrations, radiation conditions, background ozone concentration, ...).

Line 220: The authors plot the measured monthly means (noon) as a function of the FNR ratio for a large number of monitoring stations. From the summer O3 monthly means $> 160$ $\mu$g/m3 they calculate a median for the FNR (3.28). The 20% and 80%

percentiles are then used as thresholds for VOC and NOx limitation. Would the authors please comment on why it is justified to assume that thresholds can be inferred unambiguously in this way ? For this, the ozone monthly means of the measuring stations used must satisfy a given frequency distribution of the photochemical regimes. (After all, it is conceivable that the Chinese O3 monitoring network contains practically only stations with NOx limitation. Then this approach (i. e. by calculating the 50% percentile of the O3 measuring stations for which O3 monthly means > 160 $\mu$g/m3 are observed) would shift the center of the transition range far into the range of NOx limitation). Would it not be more appropriate to argue that the highest photochemical ozone production must occur in the transition region? It is therefore proposed to use an isopleth plot of summertime O3 levels above 160 $\mu$g/m3 as a function of HCHO and NO2 column densities to identify the HCHO/NO2 ratio of maximum ozone production. In this way, the HCHO/NO2 ratios for NOx and VOC limitation can be determined independently of the frequency distribution of ozone monitoring stations with values above 160 $\mu$g/m3.

Line 315: Are the same HCHO/NO2 thresholds used for the COVID-19 lockdown periods (Jan. 2010, period I and Feb. 2020 period II) from Fig. 6c and 6d as for summer conditions ? It is evident that due to the different radiation conditions alone, the ratio of directly emitted HCHO and photochemically formed HCHO (see also comment to Line 113) is significantly different in late winter than under summer conditions, so that a change in the threshold ratios for VOC- and NOx-limitation should also be expected. It is suggested that the VOC- and NOx-limitation thresholds for the COVID-19 lockdown periods should be determined independently (e.g. using the isopleth method proposed above).

Technical corrections:

Only two minor issues were found when reviewing the manuscript: - Line 68: Please replace "... to the summed rate of reactions of VOC with peroxy radicals". "... to the summed rate of reactions of VOC with OH radicals".

Line 68: The reference (Sillman, 1995) does not appear in the bibliography.

Overall, I consider the approach of the paper a promising way to describe different photochemical regimes. It will certainly be the task of further refinements of this approach in subsequent papers to make even more differentiated statements on the optimisation of ozone reduction strategies. However, satellite-based analysis of photochemical regimes does not seem to be a complete alternative to OBM studies, especially since in the first case the composition of the VOC mix (e.g. the processing of CO and CH3OH has very different HCHO production efficiencies when using the same OH reactivities for both species) is left out.

---

## Author Comment (AC1) · 21 Mar 2021

Manuscript Number: acp-2020-1097 Manuscript Name: Spatial and temporal changes of the ozone sensitivity in China based on satellite and ground-based observations

We thank the reviewers for their constructive comments and useful suggestions.

Point-by-point response to the comments: Follow your comments, our responses are written in Blue Color.

Referee #1 In their manuscript "Spatial and temporal changes of the ozone sensitiv-
ity in China based on satellite and ground-based observations", Wannan Wang et al. report on a study using OMI NO2 and HCHO columns to investigate where in China photochemical ozone production is NOx limited, and where it is VOC limited. In contrast to earlier studies using similar approaches, the classification here is based on linking satellite columns with in-situ surface ozone observations to empirically determine thresholds for the HCHO to NO2 ratio. The method is applied to data from China for the years 2016 to 2019 to investigate changes in photochemical ozone regime. Measurements from the lockdown phase in spring 2020 are included as a test case on the effect of strongly reduced NOx emissions. The manuscript is clearly written and fits well into the scope of ACP. The results are not really surprising but add information on a relevant topic in particular in view of what the best approach is to reduce ozone levels in China. I have however concerns with the basic approach taken to identify the NOx and VOC limited regimes, which is the key point of this study and not trivial. In my opinion, this needs to be better explained and justified before this manuscript can be accepted for publication in ACP.

Major comment

The one new thing in this study is that the different ozone chemistry regimes are defined from measurements alone and not from models which explicitly determine it for each location. This is an interesting approach but I do not see strong justification for it in the manuscript. What the authors use in Figure 1d is a display of monthly mean ozone surface concentrations as a function of the ratio of formaldehyde to NO2 columns. While it is tempting to look for a maximum in the ozone curve and define this as the separation between NOx and VOC sensitive domains, this is not necessarily justified. The FNR determines the change of ozone levels in reaction to a change in NOx or VOC concentrations, but not the total ozone concentration itself (as can clearly be seen in the graph). An argument can be made that by limiting the analysis to the highest values, we are indeed looking at local ozone chemistry and assuming that everything else remains unchanged, absolute ozone levels should reflect ozone production but more

justification is needed to make this approach convincing. I think that adding the threshold lines in Figure 1c may help to make the argument. Thresholds are defined from this figure in a manner not clear to me but this is of course the key question: What are the correct thresholds, are they the same throughout China, are they valid in all seasons / meteorological conditions, do they change over time as emission patterns change? None of this is discussed and this needs to be added.

Response: We have derived the FNR thresholds as follows.

We focus on the average monthly ozone surface concentrations in cities in this study to reduce the effect of meteorology or regional transport events. We consider only the highest ozone value (above 160 $\mu$g/m3) to further minimize the effect of background ozone. The margin of the top 10% ozone values per city is used to cover the transitional regime. Our thresholds are derived based on the assumption that conditions are the same throughout China, spatially and temporally. To justify our approach of deriving the FNR thresholds we add the following discussion on the assumptions:

(1) We apply FNR thresholds in different latitude zones (18°N-28°N, 28°N-38°N, 38°N-53°N) and add Figure S1 in the supplement. The results show higher FNR values in lower latitude zone which is in line with higher temperature and stronger solar radiation in lower latitude zone. The FNR thresholds in three latitude zones ([3.0, 5.5], [2.3, 4.3], [2.1, 3.9]) have overlapped. The results show the FNR thresholds [2.3, 4.2] derived for the whole domain can represent all latitude zones.

(2) We use summertime data to derive thresholds because meteorology is favorable for O3 formation and will result in high ozone concentrations far above the background. We have added Figure S2 in the supplement. Figure S2(a) shows that monthly O3 concentration in winter (Dec-Jan-Feb) will seldom exceed 160 $\mu$g/m3 and are less suitable for deriving the FNR thresholds. Based on application of FNR thresholds [2.3, 4.2] derived by summertime data to winter observations shown in Figure S2(a), it is a reasonable assumption that our observation-based FNR thresholds derived using

summertime data also apply during winter. Figure S2(b) indicates that FNR thresholds [2.3, 4.2] derived using summertime data will be valid for all seasons.

We replaced line 210 by:

"The overall O3-NO2-HCHO chemistry is also captured by satellite-based HCHO and NO2 column in Figure 1c, where we construct the O3 isopleth using only observations."

We added the discussion above in our manuscript at line 231-240:

"To minimize the effect of background O3 by transport or meteorological variability, we use monthly mean O3 concentrations above 160 $\mu$g/m3 in summer time when the O3 chemistry is strongest. We assume that the results are applicable for the whole of China. To check this assumption we investigate the FNR thresholds in different latitude zones (18°N-28°N, 28°N-38°N, 38°N-53°N) in Figure S1 in the supplement. Generally, we conclude that the derived FNR thresholds range of [2.3, 4.2] for the whole domain is a good representation for all latitude zones in China. Figure S2(a) in the supplement shows monthly O3 concentration in winter (Dec-Jan-Feb) which rarely exceed 160 $\mu$g/m3, including the FNR thresholds derived using summertime data. Based on Figure S2(b) we assume that our FNR thresholds [2.3, 4.2] derived using summertime data will be valid for all seasons."

In addition, we added the HCHO/NO2 threshold lines in Figure 1c to clarify our results.

Minor comments

At some point in the manuscript, a short discussion of the problems in using columnar data instead of surface concentrations is needed, as well as of the question, in how far monthly averages are representative of the highly variable concentrations and the strongly non-linear NOx-O3 chemistry.

Response: We agree that application of a surface-based predictor to a column-based HCHO/NO2 requires accounting for differences in the HCHO and NO2 vertical profiles as well as meteorology. Thus, we select satellite retrievals according to standards that

is considered as optimal. In addition, the monthly mean HCHO/NO2 columns are used to reduce impact of this problem. We are focusing on the average air quality related to ozone in the Chinese cities, which doesn't show extreme events, but is a useful tool for air quality management. Hence, we focus on long-term evolution in ozone sensitivity by satellite-based HCHO/NO2.

We added the following to the discussion of our manuscript at line 383-386:

"Satellite instruments measure the vertically integrated column density, which we use as a proxy of the actual surface concentrations. To reduce the effect of short-term variability in vertical distributions caused by meteorological changes we use monthly mean averages. Therefore, our satellite-based HCHO/NO2 method is limited to identification of long-term evolution in O3 sensitivity, focusing on understanding the average air quality."

L111: Add that quoted resolution for OMI is at nadir.

Response: This has been adapted.

L137: Which product is used in the DESCO algorithm – the QA4ECV product?

Response: Yes, the QA4ECV product is used for DECSO. We have added this at line 138-139: "... from OMI observations of tropospheric NO2 columns (the QA4ECV product discussed in section 2) by the Daily Emission estimation Constrained by Satellite Observations (DECSO) algorithm."

L150: Please add some basic information on which type of instrument is used for the ground-based data, which measurement principle is applied and if there are possible cross-sensitivities.

Response: The concentration of ozone and NO2 is measured by a set of continuous automated instruments. Various instruments are used. Ozone is measured by point analysers using the ultraviolet absorption spectrometry method, NO2 is measured using the chemi-luminescence method. For open path analysers, differential optical absorption spectroscopy is used to measure ozone and NO2.

We clarified this by making the following modification at line 151-155: "At each monitoring site, the concentration of O3 is measured using the ultraviolet absorption spectrometry method and differential optical absorption spectroscopy, NO2 is measured using the chemi-luminescence method. The instrumental operation, maintenance, data assurance and quality control were conducted based on the most recent revisions of China Environmental Protection Standards (CMEE, 2013)."

L168: If the model is to be used in any quantitative sense, then much more information on initialisation, VOCs used, inclusion of heterogeneous reactions etc. is needed.

Response: We agree that more information on CLASS model is needed. In this study, the model is used to show ozone formation is a highly nonlinear process in relation to NO2 and HCHO. The chemical reactions employed in model scheme are presented in van Stratum et al. (2012). VOC mainly refers to isoprene, where all its oxidation products (MVK; methyl-vinyl-ketone and MACR; methacrolein) are lumped into a single species MVK. As the scheme contains less species and equations, the computational costs are minimized while the scheme still retains the essential components of the O3-NOx-VOC-HOx cycle (Vil'a-Guerau de Arellano et al., 2011).

We added the following at line 175-177 and line 185-186:

"The initial mixing ratios of chemical species are shown in Table S1 in the supplement. The initial mixing ratios data are from van Stratum et al. (2012). All other species (except for molecular oxygen and nitrogen) are initialized at zero, and modified only the concentrations of NO2 and HCHO." "This chemical scheme is able to represent the evolution of O3-NOx-VOC-HOx cycle in semirural areas ( Vil'a-Guerau de Arellano et al., 2011; Janssen et al., 2012; van Stratum et al., 2012)."

L219: Sentence is not clear to me but probably touches on my main point of concern: That the way the thresholds are derived is oversimplified and not necessarily valid for

all locations in China.

Response: Figure 1c resembles this overall O3-NOx-VOC chemistry based on the observations, with an uncertain, blurry transition between NOx-limited and VOC-limited regimes, similar as represented in Figure 1a.

As mentioned in Response to Major comment, the FNR thresholds slightly shift from low latitude to high latitude zones. Our thresholds are derived based on statistical analysis covering more than half of samples (60%) to make it valid as much as possible in most locations in China.

We made the following modification at line 227-230:

"We find a relationship between FNR and the O3 response patterns that is qualitatively similar but quantitatively distinct across cities. Taking into account the range of transitional regime, the FNR thresholds [2.3, 4.2] marking the transitional regime, are defined as the $\pm$ 30% range from the median (3.28) covering the O3 maximum in most (60%) studied cities."

L303: Up to this point, only summer values have been discussed and used. Now, the method is suddenly used for winter values which is problematic and needs at least to be acknowledged and discussed.

Response: We use summertime data to derive thresholds because meteorology (temperature, solar radiation) is favorable for ozone formation. The local ozone maximum occurs in summer and monthly ozone value can reach above 160 $\mu$g/m3 which can be thought of as a dividing line separating two different photochemical regimes.

As mentioned in Response to Major comment, Figure S2(a) in the supplement shows monthly O3 concentration in winter (Dec-Jan-Feb) is difficult to exceed 160 $\mu$g/m3. The winter ozone observations have limited impacts on deriving the FNR thresholds. Application of FNR thresholds [2.3, 4.2] deriving by summertime data to all seasons' observations in Figure S2(b) shows it can capture the high O3 values as well. Hence,

we used the same thresholds when analyzing the COVID-19 period.

We added to the text at line 339:

"Assuming that our observation-based FNR thresholds derived using summertime data also apply during winter."

L358: Here (and earlier), the assumption Is made that VOC emissions have not decreased (much) during the lockdown which would deserve a bit of discussion, in particular as a significant number of studies related to the signature of the lockdown in pollution in China is already available in the literature.

Response: We assumed VOC emissions as "rising or in steady state" when discussed the impact of NOx emission variations on O3 concentrations in east China between 2016 and 2019 (Line 291 in original manuscript). During lockdown period, both the anthropogenic emissions of NOx and VOCs were reduced according to Sicard et al. (2020). The reductions of VOC emissions are generally effective in reducing O3 concentrations. However, such air quality improvements are largely offset by reductions in NOx emissions reductions lead to increases in O3 concentrations due to the strongly VOC-limited conditions in the North China Plain in winter (Xing et al., 2020). Figure 6c shows most regions of eastern China belong to VOC-limited conditions before lockdown. Thus, stronger NOx reduction than the VOC reduction resulted in significant O3 enhancement during lockdown.

We clarified this at line 344-348 and line 394-396:

"During lockdown period, both the anthropogenic emissions of NOx and VOCs were reduced. The NOx reduction during the lockdown is higher than the VOC reduction according to Sicard et al. (2020). The reductions of VOC emissions are generally effective in reducing O3 concentrations. However, such air quality improvements are largely offset by reductions in NOx emissions leading to increases in O3 concentrations due to the strongly VOC-limited conditions in the NCP in winter (Xing et al., 2020)."

"The case study of O3 level changes during the COVID-19 lockdown in China demonstrated that the strong reductions in anthropogenic NOx emissions resulted in significant O3 enhancement due to the VOC-limited regime in winter."

L363: I guess that something should also be said about the availability of the data produced in this study.

Response: We added the following at line 405-406:"The hourly O3 and NO2 observations of Chinese ground stations can be accessed from third parties (http://www.pm25.in, http:// www.aqicn.org)."

Figure 1: This figure needs to be revised in several ways:

1. the same colour scheme and scale should be used for model and observations;

2. the figures showing observations should not just plot the data on top of each other as this prevents readers from seeing many of them. Instead, a "heat map" type display would be more appropriate showing the binned data;

3. I'd suggest to include the threshold ratio lines also in panel 1c.

Response: We agree with the referee that it is necessary to use the same color scheme and scale for Figure 1a to 1c. Figure 1b and 1c have been adapted to "heat map". In addition, the threshold ratio lines have been added in Figure 1c.

Figure 2: I don't see the need for the inserted maps in particular as they depict disputed areas without relevance for the study. Figure 2: Add in caption that HCHO columns are from OMI.

Response: We removed the inserted maps in Figure 2. The caption of Figure 2 has been adapted.

Figure 3: Explain arrows in caption.

Response: The arrows represent time step from 2016 to 2019. This has been adapted.

Figure 5: Looking at panels (b) and (c) it is clear that the ozone changes in China are not linked in a simple way to the NOx changes: Ozone decreased very much in Neijiang in spite of very moderate NOx decreases while it increased significantly in Guangzhou where NOx emissions remained basically constant. In Beijing, a reduction in NOx emissions by a factor of 2 from 2016 to 2018 had no effect on surface ozone but further reduction then leads to an increase in O3. This needs more discussion in the text.

Response: We agree that the relationship between ozone and NOx emission is not in such a simple way. Due to different ozone formation chemistry, opposite change of ozone occurs in Guangzhou and Neijiang with NOx emission reduction. Because of VOC-limited chemistry condition, O3 increases with decreasing NOx emissions in Beijing, Shanghai and Guangzhou. NOx-limited condition leads to decreasing O3 with decreasing NOx emissions in Neijiang.

Even we assumed VOC emissions as rising or in steady state, different pollutants level and VOC species across cities would affect ozone changes in quantitative. We find a qualitative relationship between NOx emission and the O3 response patterns conforms the nonlinear O3-NO2-VOC chemistry, not in quantitative sense.

We added the following at line 318-327:

"Note that we find a qualitative relationship between NOx emission and the O3 response patterns conform the nonlinear O3-NO2-VOC chemistry, not in quantitative sense. For example, the changes of NOx emissions in Beijing (-2.17 Gg N/cell), Shanghai (-1.18 Gg N/cell), Guangzhou (-0.28 Gg N/cell) and Neijiang (-0.15 Gg N/cell) during 2016-2019 lead to different levels of O3 changes in Beijing (10.43 $\mu$g/m3), Shanghai (7.81 $\mu$g/m3), Guangzhou (25.54 $\mu$g/m3) and Neijiang (-22.66 $\mu$g/m3). Because of the VOC-limited chemistry condition, O3 increases with decreasing NOx emissions in Beijing, Shanghai and Guangzhou. The NOx-limited condition leads to decreasing O3 with decreasing NOx emissions in Neijiang. Compared with Beijing, NOx emissions in

Guangzhou remained basically constant in 2016 and 2019. But O3 concentrations in Guangzhou increased more than in Beijing. The local O3 formation sensitivity is helpful to present the way of O3 response to NOx emission, but VOC emission are needed when discussing their relationship in a quantitative way."

Figure 6: Define time periods in caption.

Response: This has been adapted.

Please also note the supplement to this comment:
https://acp.copernicus.org/preprints/acp-2020-1097/acp-2020-1097-AC1-supplement.pdf

———————————————————

[Figure]

**Fig. 1.** FNR thresholds in different latitude zones.

[Figure]

**Fig. 2.** (a) Monthly mean in-situ O3 concentrations versus NO2 columns and HCHO columns from OMI in winter (Dec-Jan-Feb) during 2016-2019.(b)same as (a), but for all season.

---

## Author Response (AR1)

**Manuscript Number:** acp-2020-1097

**Manuscript Name:** Spatial and temporal changes of the ozone sensitivity in China based on satellite and ground-based observations

**We thank the reviewers for their constructive comments and useful suggestions.**

**Point-by-point response to the comments:** Follow your comments, our responses are written in **Blue Color**.

**Referee #1**

In their manuscript "Spatial and temporal changes of the ozone sensitivity in China based on ground-based observations", Wannan Wang et al. report on a study using OMI NO2 and HCHO columns to investigate where in China photochemical ozone production is NOx limited, and where it is VOC limited. In contrast to earlier studies using similar approaches, the classification here is based on linking satellite columns with in-situ surface ozone observations to empirically determine thresholds for the HCHO to NO2 ratio. The method is applied to data from China for the years 2016 to 2019 to investigate changes in photochemical ozone regime. Measurements from the lockdown phase in spring 2020 are included as a test case on the effect of strongly reduced NOx emissions.

The manuscript is clearly written and fits well into the scope of ACP. The results are not really surprising but add information on a relevant topic in particular in view of what the best approach is to reduce ozone levels in China. I have however concerns with the basic approach taken to identify the NOx and VOC limited regimes, which is the key point of this study and not trivial. In my opinion, this needs to be better explained and justified before this manuscript can be accepted for publication in ACP.

**Major comment**

The one new thing in this study is that the different ozone chemistry regimes are defined from measurements alone and not from models which explicitly determine it for each location. This is an interesting approach but I do not see strong justification for it in the manuscript. What the authors use in Figure 1d is a display of monthly mean ozone surface concentrations as a function of the ratio of formaldehyde to NO2 columns. While it is tempting to look for a maximum in the ozone curve and define this as the separation between NOx and VOC sensitive domains, this is not necessarily justified. The FNR determines the change of ozone levels in reaction to a change in NOx or VOC concentrations, but not the total ozone concentration itself (as can clearly be seen in the graph). An argument can be made that by limiting the analysis to the highest values, we are indeed looking at local ozone chemistry and

assuming that everything else remains unchanged, absolute ozone levels should reflect ozone production but more justification is needed to make this approach convincing. I think that adding the threshold lines in Figure 1c may help to make the argument. Thresholds are defined from this figure in a manner not clear to me but this is of course the key question: What are the correct thresholds, are they the same throughout China, are they valid in all seasons / meteorological conditions, do they change over time as emission patterns change? None of this is discussed and this needs to be added.

**Response:** We have derived the FNR thresholds as follows.

We focus on the average monthly ozone surface concentrations in cities in this study to reduce the effect of meteorology or regional transport events. We consider only the highest ozone value (above 160 μg/m$^3$) to further minimize the effect of background ozone. The margin of the top 10% ozone values per city is used to cover the transitional regime. Our thresholds are derived based on the assumption that conditions are the same throughout China, spatially and temporally. To justify our approach of deriving the FNR thresholds we add the following discussion on the assumptions:

(1) We apply FNR thresholds in different latitude zones (18 N-28 N, 28 N-38 N, 38 N-53 N) and add Figure S1 in the supplement. The results show higher FNR values in lower latitude zone which is in line with higher temperature and stronger solar radiation in lower latitude zone. The FNR thresholds in three latitude zones ([3.0, 5.5], [2.3, 4.3], [2.1, 3.9]) have overlapped. The results show the FNR thresholds [2.3, 4.2] derived for the whole domain can represent all latitude zones.

(2) We use summertime data to derive thresholds because meteorology is favorable for O$_3$ formation and will result in high ozone concentrations far above the background. We have added Figure S2 in the supplement. Figure S2(a) shows that monthly O$_3$ concentration in winter (Dec-Jan-Feb) will seldom exceed 160 μg/m$^3$ and are less suitable for deriving the FNR thresholds. Based on application of FNR thresholds [2.3, 4.2] derived by summertime data to winter observations shown in Figure S2(a), it is a reasonable assumption that our observation-based FNR thresholds derived using summertime data also apply during winter. Figure S2(b) indicates that FNR thresholds [2.3, 4.2] derived using summertime data will be valid for all seasons.

[Figure]

Figure S1. (a) Derived FNR thresholds in 8 °N-28 °N. (b) same as (a), but in 28 °N-38 °N. (c) same as (a), but in 38 °N-53 °N. (d) The monthly mean in-situ $O_3$ concentrations versus $NO_2$ columns and HCHO columns from OMI in 8 °N-28 °N. (e) same as (d), but in 28 °N-38 °N. (e) same as (d), but in 38 °N-53 °N.

[Figure]

Figure S2. (a) The 360 cities' monthly mean in-situ $O_3$ concentrations versus $NO_2$ columns and HCHO columns from OMI observations in winter (Dec-Jan-Feb) during 2016-2019. (b) same as (a), but for all seasons.

We replaced line 210 by:

*"The overall $O_3$-$NO_2$-HCHO chemistry is also captured by satellite-based HCHO and $NO_2$ column in Figure 1c, where we construct the $O_3$ isopleth using only observations."*

We added the discussion above in our manuscript at line 231-240:

*"To minimize the effect of background $O_3$ by transport or meteorological variability, we use monthly mean $O_3$ concentrations above 160 $\mu g/m^3$ in summer time when the $O_3$ chemistry is strongest. We assume that the results are applicable for the whole of China. To check this assumption we investigate the FNR thresholds in different latitude zones (18 N-28 N, 28 N-38 N, 38 N-53 N) in Figure S1 in the supplement. Generally, we conclude that the derived FNR thresholds range of [2.3, 4.2] for the whole domain is a good representation for all latitude zones in China.*

*Figure S2(a) in the supplement shows monthly $O_3$ concentration in winter (Dec-Jan-Feb) which rarely exceed 160 $\mu g/m^3$, including the FNR thresholds derived using summertime data. Based on Figure S2(b) we assume that our FNR thresholds [2.3, 4.2] derived using summertime data will be valid for all seasons."*

In addition, we added the HCHO/$NO_2$ threshold lines in Figure 1c to clarify our results.

**Minor comments**

At some point in the manuscript, a short discussion of the problems in using columnar data instead of surface concentrations is needed, as well as of the question, in how far monthly averages are representative of the highly variable concentrations and the strongly non-linear NOx-O3 chemistry.

**Response:** We agree that application of a surface-based predictor to a column-based $HCHO/NO_2$ requires accounting for differences in the HCHO and $NO_2$ vertical profiles as well as meteorology. Thus, we select satellite retrievals according to standards that is considered as optimal. In addition, the monthly mean $HCHO/NO_2$ columns are used to reduce impact of this problem. We are focusing on the average air quality related to ozone in the Chinese cities, which doesn't show extreme events, but is a useful tool for air quality management. Hence, we focus on long-term evolution in ozone sensitivity by satellite-based $HCHO/NO_2$.

We added the following to the discussion of our manuscript at line 383-386:

*"Satellite instruments measure the vertically integrated column density, which we use as a proxy of the actual surface concentrations. To reduce the effect of short-term variability in vertical distributions caused by meteorological changes we use monthly mean averages. Therefore, our satellite-based $HCHO/NO_2$ method is limited to identification of long-term evolution in $O_3$ sensitivity, focusing on understanding the average air quality."*

L111: Add that quoted resolution for OMI is at nadir.

**Response:** This has been adapted.

L137: Which product is used in the DESCO algorithm – the QA4ECV product?

**Response:** Yes, the QA4ECV product is used for DECSO. We have added this at line 138-139:

*"… from OMI observations of tropospheric $NO_2$ columns (the QA4ECV product discussed in section 2) by the Daily Emission estimation Constrained by Satellite Observations (DECSO) algorithm."*

L150: Please add some basic information on which type of instrument is used for the ground-based data, which measurement principle is applied and if there are possible cross-sensitivities.

**Response:** The concentration of ozone and $NO_2$ is measured by a set of continuous automated instruments. Various instruments are used. Ozone is measured by point analysers using the ultraviolet absorption spectrometry method, $NO_2$ is measured using the chemi-luminescence method. For open path analysers, differential optical absorption spectroscopy is used to measure ozone and $NO_2$.

We clarified this by making the following modification at line 151-155:

*"At each monitoring site, the concentration of O₃ is measured using the ultraviolet absorption spectrometry method and differential optical absorption spectroscopy, NO₂ is measured using the chemi-luminescence method. The instrumental operation, maintenance, data assurance and quality control were conducted based on the most recent revisions of China Environmental Protection Standards (CMEE, 2013)."*

L168: If the model is to be used in any quantitative sense, then much more information on initialisation, VOCs used, inclusion of heterogeneous reactions etc. is needed.

**Response:** We agree that more information on CLASS model is needed. In this study, the model is used to show ozone formation is a highly nonlinear process in relation to $NO_2$ and HCHO. The chemical reactions employed in model scheme are presented in van Stratum et al. (2012). VOC mainly refers to isoprene, where all its oxidation products (MVK; methyl-vinyl-ketone and MACR; methacrolein) are lumped into a single species MVK. As the scheme contains less species and equations, the computational costs are minimized while the scheme still retains the essential components of the $O_3$-$NO_x$-VOC-$HO_x$ cycle (Vil`a-Guerau de Arellano et al., 2011).

We added the following at line 175-177 and line 185-186:

*"The initial mixing ratios of chemical species are shown in Table S1 in the supplement. The initial mixing ratios data are from van Stratum et al. (2012). All other species (except for molecular oxygen and nitrogen) are initialized at zero, and modified only the concentrations of NO₂ and HCHO."*

*"This chemical scheme is able to represent the evolution of O₃-NOₓ-VOC-HOₓ cycle in semirural areas ( Vil`a-Guerau de Arellano et al., 2011; Janssen et al., 2012; van Stratum et al., 2012)."*

*Table S1. Initial mixing ratio in the mixed-layer.*

| Species | $O_3$ | NO | $CH_4$ | isoprene | CO | $H_2O_2$ |
|---|---|---|---|---|---|---|
| Units (ppb) | 30 | 0 | 1724 | 0 | 105 | 0.1 |

L219: Sentence is not clear to me but probably touches on my main point of concern: That the way the thresholds are derived is oversimplified and not necessarily valid for all locations in China.

**Response:** Figure 1c resembles this overall $O_3$-$NO_x$-VOC chemistry based on the observations, with an uncertain, blurry transition between $NO_x$-limited and VOC-limited regimes, similar as represented in Figure 1a.

As mentioned in Response to Major comment, the FNR thresholds slightly shift from low latitude to high latitude zones. Our thresholds are derived based on statistical analysis covering more than half of samples (60%) to make it valid as much as possible in most locations in China.

We made the following modification at line 227-230:

*"We find a relationship between FNR and the $O_3$ response patterns that is qualitatively similar but quantitatively distinct across cities. Taking into account the range of transitional regime, the FNR thresholds [2.3, 4.2] marking the transitional regime, are defined as the ±30% range from the median (3.28) covering the $O_3$ maximum in most (60%) studied cities."*

L303: Up to this point, only summer values have been discussed and used. Now, the method is suddenly used for winter values which is problematic and needs at least to be acknowledged and discussed.

**Response:** We use summertime data to derive thresholds because meteorology (temperature, solar radiation) is favorable for ozone formation. The local ozone maximum occurs in summer and monthly ozone value can reach above 160 μg/m$^3$ which can be thought of as a dividing line separating two different photochemical regimes.

As mentioned in Response to Major comment, Figure S2(a) in the supplement shows monthly $O_3$ concentration in winter (Dec-Jan-Feb) is difficult to exceed 160 μg/m$^3$. The winter ozone observations have limited impacts on deriving the FNR thresholds. Application of FNR thresholds [2.3, 4.2] deriving by summertime data to all seasons' observations in Figure S2(b) shows it can capture the high $O_3$ values as well. Hence, we used the same thresholds when analyzing the COVID-19 period.

We added to the text at line 339:

*"Assuming that our observation-based FNR thresholds derived using summertime data also apply during winter."*

L358: Here (and earlier), the assumption Is made that VOC emissions have not decreased (much) during the lockdown which would deserve a bit of discussion, in particular as a significant number of studies related to the signature of the lockdown in pollution in China is already available in the literature.

**Response:** We assumed VOC emissions as "rising or in steady state" when discussed the impact of NO$_x$ emission variations on $O_3$ concentrations in east China between 2016 and 2019 (Line 291 in original manuscript). During lockdown period, both the anthropogenic emissions of NO$_x$ and VOCs were reduced according to Sicard et al. (2020). The reductions of VOC emissions are generally effective in reducing $O_3$ concentrations. However, such air quality improvements are largely offset by reductions in NO$_x$ emissions reductions lead to increases in $O_3$

concentrations due to the strongly VOC-limited conditions in the North China Plain in winter (Xing et al., 2020). Figure 6c shows most regions of eastern China belong to VOC-limited conditions before lockdown. Thus, stronger $NO_x$ reduction than the VOC reduction resulted in significant $O_3$ enhancement during lockdown.

We clarified this at line 344-348 and line 394-396:

*"During lockdown period, both the anthropogenic emissions of $NO_x$ and VOCs were reduced. The $NO_x$ reduction during the lockdown is higher than the VOC reduction according to Sicard et al. (2020). The reductions of VOC emissions are generally effective in reducing $O_3$ concentrations. However, such air quality improvements are largely offset by reductions in $NO_x$ emissions leading to increases in $O_3$ concentrations due to the strongly VOC-limited conditions in the NCP in winter (Xing et al., 2020)."*

*"The case study of $O_3$ level changes during the COVID-19 lockdown in China demonstrated that the strong reductions in anthropogenic $NO_x$ emissions resulted in significant $O_3$ enhancement due to the VOC-limited regime in winter."*

L363: I guess that something should also be said about the availability of the data produced in this study.

**Response:** We added the following at line 405-406:

*"The hourly $O_3$ and $NO_2$ observations of Chinese ground stations can be accessed from third parties (http://www.pm25.in, http:// www.aqicn.org)."*

Figure 1: This figure needs to be revised in several ways:

1. the same colour scheme and scale should be used for model and observations;

2. the figures showing observations should not just plot the data on top of each other as this prevents readers from seeing many of them. Instead, a "heat map" type display would be more appropriate showing the binned data;

3. I'd suggest to include the threshold ratio lines also in panel 1c.

**Response:** We agree with the referee that it is necessary to use the same color scheme and scale for Figure 1a to 1c. Figure 1b and 1c have been adapted to "heat map". In addition, the threshold ratio lines have been added in Figure 1c.

Figure 2: I don't see the need for the inserted maps in particular as they depict disputed areas without relevance for the study.

Figure 2: Add in caption that HCHO columns are from OMI.

**Response:** We removed the inserted maps in Figure 2. The caption of Figure 2 has been adapted.

Figure 3: Explain arrows in caption.

**Response:** The arrows represent time step from 2016 to 2019. This has been adapted.

Figure 5: Looking at panels (b) and (c) it is clear that the ozone changes in China are not linked in a simple way to the NOx changes: Ozone decreased very much in Neijiang in spite of very moderate NOx decreases while it increased significantly in Guangzhou where NOx emissions remained basically constant. In Beijing, a reduction in NOx emissions by a factor of 2 from 2016 to 2018 had no effect on surface ozone but further reduction then leads to an increase in O3. This needs more discussion in the text.

**Response:** We agree that the relationship between ozone and $NO_x$ emission is not in such a simple way. Due to different ozone formation chemistry, opposite change of ozone occurs in Guangzhou and Neijiang with $NO_x$ emission reduction. Because of VOC-limited chemistry condition, $O_3$ increases with decreasing $NO_x$ emissions in Beijing, Shanghai and Guangzhou. $NO_x$-limited condition leads to decreasing $O_3$ with decreasing $NO_x$ emissions in Neijiang.

Even we assumed VOC emissions as rising or in steady state, different pollutants level and VOC species across cities would affect ozone changes in quantitative. We find a qualitative relationship between $NO_x$ emission and the $O_3$ response patterns conforms the nonlinear $O_3$-$NO_2$-VOC chemistry, not in quantitative sense.

We added the following at line 318-327:

*"Note that we find a qualitative relationship between $NO_x$ emission and the $O_3$ response patterns conform the nonlinear $O_3$-$NO_2$-VOC chemistry, not in quantitative sense. For example, the changes of $NO_x$ emissions in Beijing (-2.17 Gg N/cell), Shanghai (-1.18 Gg N/cell), Guangzhou (-0.28 Gg N/cell) and Neijiang (-0.15 Gg N/cell) during 2016-2019 lead to different levels of $O_3$ changes in Beijing (10.43 $\mu g/m^3$), Shanghai (7.81 $\mu g/m^3$), Guangzhou (25.54 $\mu g/m^3$) and Neijiang (-22.66 $\mu g/m^3$). Because of the VOC-limited chemistry condition, $O_3$ increases with decreasing $NO_x$ emissions in Beijing, Shanghai and Guangzhou. The $NO_x$-limited condition leads to decreasing $O_3$ with decreasing $NO_x$ emissions in Neijiang. Compared with Beijing, $NO_x$ emissions in Guangzhou remained basically constant in 2016 and 2019. But $O_3$ concentrations in Guangzhou increased more than in Beijing. The local $O_3$ formation sensitivity is helpful to present the way of $O_3$ response to $NO_x$ emission, but VOC emission are needed when discussing their relationship in a quantitative way."*

Figure 6: Define time periods in caption.

**Response:** This has been adapted.

**Referee #2**

**General comments:**

The paper provides a substantial contribution to the understanding of photochemical ozone production in China. Instead of looking directly at ozone precursors (NOx and VOCs) in the field, it is demonstrated here that information on the photochemical regime can be derived from satellite data of NO2 and HCHO column densities. In this context, the authors develop a method to discriminate between VOC and NOX dependence using characteristic monthly HCHO/NO2 column density ratios from satellite observations. An important aspect of this approach is that they contrast satellite-based HCHO/NO2 ratios for a large number of ground-based ozone measurements and derive characteristic thresholds for VOC or NOx dependence of ozone formation. In the last part of the paper, the developed methodology is applied to the exceptional situation of the COVID-19 lockdown in late winter 2020 in China. The increase in ozone concentrations observed for numerous areas in eastern China was contrasted with the parallel observed decrease in NO2 column densities, and resulting changes for the pre/post lockdown photochemical regimes prevailing in China were derived from the satellite-based HCHO/NO2 ratios. The authors conclude by pointing out that China's future ozone reduction strategy should in any case be accompanied by a parallel VOC reduction control in addition to the focus on NOx reductions that has prevailed so far. The paper is well written and structured and clearly identifies the scientific sources used. The authors make a clear distinction between their own contributions and adequately acknowledge previous work from the literature. Overall, the following ratings are given: Scientific significance: Excellent (4) Scientific quality: Good (3) (see comments) Presentation quality: Excellent (4)

**Special comments:**

Line 113: The authors point out that only data from the afternoon are used to determine the HCHO/NO2 ratios in order to describe the peak of photochemical ozone production. However, HCHO is a trace gas that is both emitted and photochemically formed in parallel with ozone production. Would the authors please comment on why they do not distinguish between directly emitted HCHO and photochemically formed HCHO. Shouldn't only photochemically formed HCHO be a measure of the intensity of ozone formation? And would it not be possible to draw conclusions about the proportion of directly emitted HCHO (e.g. from vehicle exhaust gases) from additional "satellite winter data" to be evaluated?

**Response:** We presented the level of ozone formed from photo-oxidation of total measured HCHO only, not differentiating the contributions from different sources. HCHO (no matter from directly emitted or from secondary formation) as an important precursor of ozone, the contribution to ozone formation was evaluated with *in situ* observations of ozone in our study. Previous modeling studies derive the thresholds by simulating the response of surface ozone to an overall reduction in $NO_x$ or NMVOC emissions with coarse resolution models, which best capture regional as opposed to local $O_3$-$NO_x$-VOC sensitivity. Our thresholds derived with *in situ* observations should be more indicative of the local ozone chemistry, including HCHO photochemical formation and the effect of $NO_x$ titration over urban areas.

HCHO exhibits significant seasonal variations: the concentrations observed in autumn and summer were higher than those in winter and spring. Due to the higher temperature and stronger solar radiation in summer, the higher concentration level of HCHO mainly results from the intense photo-oxidation of VOCs, while direct anthropogenic emissions (e.g. vehicle exhaust gases) may be more important in winter. However, the discrepancy between summer and winter is not prominent in all regions, which may be associated with the relatively smaller temperature difference and the unfavorable vertical diffusion conditions in winter. For example, in Shenzhen (a typical city located in the Pearl River Delta), the ambient daytime HCHO is dominated by anthropogenic secondary formation (about 39%) in all seasons, while anthropogenic primary sources contributed the most during the winter (18%) and spring (20%) in terms of seasonal variation (Wang et al., 2017).

We added the following to the discussion of our manuscript at line 387-389:

*"We presented the level of $O_3$ formed from photo-oxidation of total measured HCHO only, not differentiating the contributions from different sources (directly emitted or photochemical formed). Due to the higher temperature and stronger solar radiation in summer, the higher concentration level of HCHO mainly results from the intense photo-oxidation of VOCs."*

Line 120: The authors point out that they use solar zenith angles of < 80 to determine the monthly HCHO/NO2 column density ratios. However, due to the finite pixel size, the intensity of HCHO production depends on the integral solar radiation lasting for several hours. Therefore, should not different HCHO/NO2 ratios be used due to the enormous extent of the study area (< 20_N - > 45_N)?

**Response:** All satellite data is used in this study. However certain filters had to applied to select qualitatively good data. These criteria will not affect the amount of data very much.

We added the following at line 120:

*"We select QA4ECV NO$_2$ daily observations following the recommendations given in the Product Specification Document (Boersma et al., 2011) for this data product."*

Line 129: The authors point out that the monthly mean values of the HCHO column densities of (0.05 * 0.05) have been converted to the pixel size for NO2 of (0.125*0.125). Can it be excluded that this procedure leads to significant changes in the HCHO/NO2 ratios (after all, the ratio of the column densities of HCHO/NO2 contains the quotient of the precursor (NO2) and the product of photochemical processing. The adaptation of the HCHO pixel size to that of NO2 could lead to a systematic underestimation of the HCHO column densities. Would the authors please comment on this?

**Response:** Satellite-based HCHO and NO$_2$ products used in this study have the same original pixel size. Because they are measured by the same instrument OMI, but they are disseminated in different grids. It is fair to make the resolution of both products the same. We do not intent to present analysis on street-level, but we show results for averages on city-level. For this the resolution of 0.125*0.125 degree is good match.

To avoid systematic impact caused by the division operation we only consider grids of monthly HCHO columns higher than $2 \times 10^{15}$ molecule/cm$^2$ (detection limitation) and NO$_2$ columns more than $1.5 \times 10^{15}$ molecule/cm$^2$ (which are defined as polluted regions).

Line 175: Using the CLASS model, the authors demonstrate that photochemical ozone production can be represented in terms of O3 isopleths as a function of HCHO (as a proxi for VOC) and NO2. Shouldn't the isopleths rather represent the ozone production over a period of time instead of the total ozone concentrations shown? Elsewhere, the authors explicitly point out that they distinguish between background ozone and additional ozone production by local photochemical ozone production in the measured ozone monthly means (c. f. line 207). It is suggested to describe the non-linearity of ozone production either only schematically (in the form of a cartoon) or actually quantitatively by naming all boundary conditions (starting concentrations, radiation conditions, background ozone concentration, ...).

**Response:** We use a box-model to simulate the real situation in cities including background level of ozone. Our model results are indeed limited and that is why we use monthly mean data to average out transport. But we still include background ozone. For the isopleths in Figure 1c we used monthly average values of O$_3$, HCHO and NO$_2$. We present atmosphere dynamic and chemistry conditions adopted in model in Table 1. We added the initial mixing ratios of chemical species are shown in Table S1 in the supplement.

We added the following at 175-177 and line 185-186:

*"The initial mixing ratios of chemical species are shown in Table S1 in the supplement. The initial mixing ratios data are from van Stratum et al. (2012). All other species (except for molecular oxygen and nitrogen) are initialized at zero, and modified only the concentrations of $NO_2$ and HCHO."*

*"This chemical scheme is able to represent the evolution of $O_3$-$NO_x$-VOC-$HO_x$ cycle in semirural areas ( Vil`a-Guerau de Arellano et al., 2011; Janssen et al., 2012; van Stratum et al., 2012)."*

*Table S1. Initial mixing ratio in the mixed-layer.*

| Species | $O_3$ | NO | $CH_4$ | isoprene | CO | $H_2O_2$ |
|---|---|---|---|---|---|---|
| Units (ppb) | 30 | 0 | 1724 | 0 | 105 | 0.1 |

Line 220: The authors plot the measured monthly means (noon) as a function of the FNR ratio for a large number of monitoring stations. From the summer O3 monthly means > 160 ug/m3 they calculate a median for the FNR (3.28). The 20% and 80% percentiles are then used as thresholds for VOC and NOx limitation. Would the authors please comment on why it is justified to assume that thresholds can be inferred unambiguously in this way? For this, the ozone monthly means of the measuring stations used must satisfy a given frequency distribution of the photochemical regimes. (After all, it is conceivable that the Chinese O3 monitoring network contains practically only stations with NOx limitation. Then this approach (i. e. by calculating the 50% percentile of the O3 measuring stations for which O3 monthly means > 160 ug/m3 are observed) would shift the center of the transition range far into the range of NOx limitation). Would it not be more appropriate to argue that the highest photochemical ozone production must occur in the transition region? It is therefore proposed to use an isopleth plot of summertime O3 levels above 160 ug/m3 as a function of HCHO and NO2 column densities to identify the HCHO/NO2 ratio of maximum ozone production. In this way, the HCHO/NO2 ratios for NOx and VOC limitation can be determined independently of the frequency distribution of ozone monitoring stations with values above 160 ug/m3.

**Response:** The suggestion of the reviewer is actually exactly the approach we have applied and it is shown in Figure 1c. Apparently that has not been made clear in the original text. We have changed the explanation to the following at line 217-223:

*"We calculated for each city the monthly mean surface $O_3$ as function of the monthly column densities of $NO_2$ and HCHO for all months during May – October from 2016 to 2019. The results are shown in Figure 1c. We only consider observations of monthly HCHO columns higher than $2 \times 10^{15}$ molecule/cm$^2$ (detection limitation), $NO_2$ columns more than $1.5 \times 10^{15}$ molecule/cm$^2$ (which are defined as polluted regions) and $O_3$ columns above 160 $\mu g/m^3$ (minimizing the effect of background ozone). We then plot in Figure 1d the surface $O_3$ concentrations as*

*function of the FNR to determine the range of FNRs, which includes the $O_3$ maximum for most (> 60%) cities. This range, which we define as the transition between the $NO_x$-limited and VOC-limited regime.*"

Line 315: Are the same HCHO/NO2 thresholds used for the COVID-19 lockdown periods (Jan. 2010, period I and Feb. 2020 period II) from Fig. 6c and 6d as for summer conditions? It is evident that due to the different radiation conditions alone, the ratio of directly emitted HCHO and photochemically formed HCHO (see also comment to Line 113) is significantly different in late winter than under summer conditions, so that a change in the threshold ratios for VOC- and NOx-limitation should also be expected. It is suggested that the VOC- and NOx-limitation thresholds for the COVID-19 lockdown periods should be determined independently (e.g. using the isopleth method proposed above).

**Response:** We agree directly emitted HCHO might be more important in winter. But the contribution of HCHO to $O_3$ formation was evaluated with *in situ* observations of $O_3$ in our study which including HCHO photochemical formation and the effect of $NO_x$ titration in the local ozone chemistry.

We use summertime data to derive thresholds because meteorology conditions (temperature, solar radiation) are favorable for ozone formation.

In Figure S2(a) in the supplement we show monthly $O_3$ concentration in winter (Dec-Jan-Feb) as function of $NO_2$ and HCHO. It is much harder to derive the FNR thresholds from these values. We assume the thresholds derive from the summertime data can be applied in winter too.

Application of FNR thresholds [2.3, 4.2] derived by summertime data to winter observations in Figure S2(a) shows it is a reasonable assumption. Hence, we used the same thresholds when analyzing the COVID-19 period.

We added the discussion above in our manuscript at line 236-238:

*"Figure S2(a) in the supplement shows monthly $O_3$ concentration in winter (Dec-Jan-Feb) which rarely exceed 160 $\mu g/m^3$, including the FNR thresholds derived using summertime data. Based on Figure S2(b) we assume that our FNR thresholds [2.3, 4.2] derived using summertime data will be valid for all seasons."*

**Technical corrections:**

Only two minor issues were found when reviewing the manuscript: - Line 68: Please replace "... to the summed rate of reactions of VOC with peroxy radicals". "... to the summed rate of reactions of VOC with OH radicals".

Line 68: The reference (Sillman, 1995) does not appear in the bibliography.

**Response:** This reference has been added.

Overall, I consider the approach of the paper a promising way to describe different photochemical regimes. It will certainly be the task of further refinements of this approach in subsequent papers to make even more differentiated statements on the optimization of ozone reduction strategies. However, satellite-based analysis of photochemical regimes does not seem to be a complete alternative to OBM studies, especially since in the first case the composition of the VOC mix (e.g. the processing of CO and CH3OH has very different HCHO production efficiencies when using the same OH reactivities for both species) is left out.